# High harmonic spectroscopy reveals anisotropy of the charge-density-wave phase transition in TiSe$_2$
Igor Tyulnev[1], Lin Zhang [1], Lenard Vamos[1], Julita Poborska[1], Utso Bhattacharya[2],
Ravindra W. Chhajlany[3], Tobias Grass [4,5], Samuel Mañas-Valero [6], Eugenio Coronado[6],
Maciej Lewenstein [1,7] & Jens Biegert [1,7] ✉

Charge density waves (CDW) appear as periodic lattice deformations which arise from electron-phonon and excitonic correlations and provide a path towards the study of condensate phases at high temperatures. While characterization of this correlated phase is well established via real or reciprocal space techniques, for systems where the mechanisms interplay, a macroscopic approach becomes necessary. Here, we demonstrate the application of polarization-resolved high-harmonic generation (HHG) spectroscopy to investigate the correlated CDW phase and transitions in TiSe$_2$. Unlike previous studies focusing on static crystallographic properties, the research examines the dynamic reordering that occurs within the CDW as the material is cooled from room temperature to 14 K. By linking ultrafast field-driven dynamics to the material's potential landscape, the study demonstrates HHG's unique sensitivity to highly correlated phases and their strength. The findings reveal an anisotropic component below the CDW transition temperature, providing insights into the nature of this phase. The investigation highlights the interplay between linear and nonlinear optical responses and their departure from simple perturbative dynamics, offering a fresh perspective on correlated quantum phases in condensed matter systems.

Highly correlated quantum phases emerge from many-body interactions between charge carriers and the lattice, enabling quantum phenomena to manifest at macroscopic scales[1]. In systems like 1T-TiSe$_2$, these interactions facilitate the formation of bosonic quasiparticles via phonon mediation, much like Cooper pair formation in superconductivity[2,3]. A related phase, the charge density wave (CDW)[4,5], follows similar principles, where the lattice vector Q connects high-symmetry points in the Brillouin zone, enabling exciton formation and a $2 \times 2 \times 2$ commensurate periodic lattice distortion below ~200 K[6–10]. Recent studies[11–13] provide evidence for exciton condensation[14–16] in TiSe$_2$ when the exciton binding energy exceeds the bandgap, pointing to a novel phase of matter. Moreover, the material exhibits superconductivity upon copper intercalation[17] or applied pressure[18], making it central to exploring high-temperature condensates. Anomalies such as chiral CDW stabilization[19,20], anisotropy[21], and strong CDW responses[6] add to the ongoing debate over the CDW mechanism—whether driven by electron-phonon interactions[9,22–24] or excitonic effects[11,25,26]. In this context, both electron-phonon (Jahn-Teller) and excitonic mechanisms have been proposed, and each aligns with experimental observations. Using high-harmonic generation spectroscopy, we investigate this prototypical phase transition from a different perspective, as it probes changes in correlations and symmetries[27,28].

## Results

### High harmonic spectroscopy

At the core of high harmonic generation (HHG) in solids[29–31] lies the precise response to charge currents through polarization and by crystal orientation. Analogous to the three-step model in gaseous HHG, charge carriers in solids are excited by strong optical fields, predominantly at bandgap minima, and subsequently accelerated. The trajectories of these carriers, governed by field polarization and atomic potentials, allow recombination at the origin and at

[1]ICFO—Institut de Ciencies Fotoniques, The Barcelona Institute of Science and Technology, Barcelona, Spain. [2]Institute for Theoretical Physics, ETH Zurich, Zurich, Switzerland. [3]ISQI—Institute of Spintronics and Quantum Information, Faculty of Physics and Astronomy, Adam Mickiewicz University, Poznań, Poland. [4]DIPC—Donostia International Physics Center, Paseo Manuel de Lardizábal 4, San Sebastián, Spain. [5]Ikerbasque—Basque Foundation for Science, Plaza Euskadi 5, Bilbao, Spain. [6]Instituto de Ciencia Molecular (ICMol), Universitat de València, Paterna, Spain. [7]ICREA, Pg. Lluís Companys 23, Barcelona, Spain. ✉e-mail: jens.biegert@icfo.eu

other lattice sites, enabling HHG spectroscopy to resolve crystallographic details directly[32,33]. The harmonic response, linked to optical properties and the complex dielectric function, is approximated through the Drude-Lorentz oscillator model, where resonances arise from interband transitions and their electron-hole populations.

Higher harmonics are driven at moderate intensities using the pon-deromotive scaling for HHG cut-off energy to preserve the correlated phase. We employ a mid-infrared optical parametric chirp pulse amplification (OPCPA) system[34,35], providing an optical field centered at 3.2 μm. This driving field impinges onto the sample in reflection geometry at a 45° incidence, and the reflected signal is spectrally analyzed. Harmonics are driven up to the 7th order for field amplitudes reaching $0.078 \pm 0.007\ V\,A^{-1}$, while the linear response is captured through reflectivity measurements. Note that we quote field amplitudes outside the material since the dielectric constant changes with temperature.

Reflectivity data exhibit a phase-dependent response, remaining constant above the transition temperature ($T_c$) with quasi-linear scaling below $T_c$. A linear fit yields the transition temperature, resulting in $T_c = 197.3 \pm 3.5$ K, in agreement with the literature[6,7]. The resemblance to the CDW gap scaling becomes clear when examining the changes in optical properties near the photon energy of our fundamental field. As shown in Fig. 1c, the reflectivity is entirely governed by the temperature-dependent variations in optical conductivity or the dielectric function (see Supplementary Table 1). In photoemission experiments, the CDW gap is typically inferred from measured shifts in the valence band ($v_1$) relative to the static spectator conduction bands ($c_1$ and $c_2$), which emerge during back folding below $T_c$; see Fig. 1a, b. This region is significantly affected by changes in the chemical potential. It contributes to the low-frequency response, including the appearance of a plasma edge and phonon peaks; shown in Fig. 1c. In contrast, the larger gap between $v_1$ and $c_3$ is readily accessed by photon-based measurements. Here, we probe the $v_1$-$c_3$ gap with a laser at a fundamental frequency $\omega_0 = 0.39\,eV$.

Measuring the optical conductivity and reflectivity of $TiSe_2$ across the gap thus directly probes the band renormalization and CDW gap modified by the temperature-dependent scattering rate. Figure 1d shows the result of these measurements and reveals up to 22% relative rise in reflectivity at the lowest measured temperature of 14 K.

Now, we turn to exploring the harmonic yields as a function of temperature. To gauge the regime of interaction, we conduct a power scan (see Supplementary Fig. 2). We find the below-bandgap harmonics H3 and H5 scale perturbatively, while the above-gap H7 switches over to non-perturbative scaling. Figure 2 reveals stark differences in the behavior of various harmonic orders, particularly when the driving field polarization aligns with the Γ-K direction. Approximating the expected material response with the perturbative model for harmonic generation[36], we expect that changes in harmonic yield reflect the temperature-dependent variations in nonlinear susceptibilities for harmonic orders. Based on Miller's rule, we express the nonlinear susceptibility as a product of linear susceptibilities at harmonic frequencies with the fundamental susceptibility raised to the respective power $\chi^{(N)} \propto \chi^{(1)}(N \cdot \omega_0)\left[\chi^{(1)}(\omega_0)\right]^N$. Previous broadband studies (Fig. 1c) show that the optical response at harmonic frequencies is nearly independent of temperature. This suggests that harmonic generation is primarily influenced by changes at the fundamental frequency, where the resonance appears.

Using our mean-field model, we calculate the temperature-dependent linear susceptibility along the Γ-K direction at 0.4 eV (Fig. 2b). The results show a strong sensitivity to the scattering rate, particularly at low temperatures. Raising this susceptibility to the 3rd, 5th, and 7th powers, we find that the behavior of H3 and H7 in Fig. 2a matches well with the model, especially at a scattering rate of 5 meV. However, H5 deviates significantly from this trend, showing a pronounced minimum around 80 K.

Exploring this further along the Γ-M directions (Fig. 2c), H5 again behaves differently, despite the Γ-$M_1$ and Γ-$M_2$ directions being theoretically

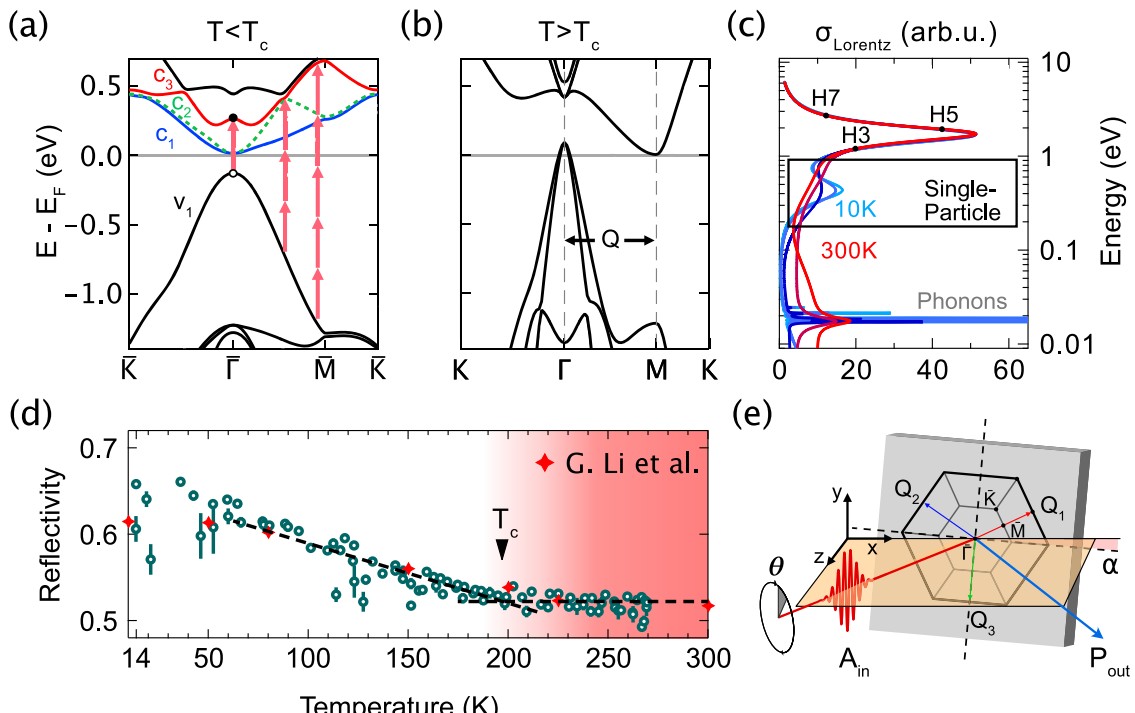

**Fig. 1 | Band structures and Drude Lorentz model. a, b** Reduced and normal band structure of 1T-TiSe$_2$. The lattice vector Q, connecting the Γ and M points halves the Brillouin zone, leading to band back-folding and renormalization below transition temperature $T_c$. The shifting of valence band $v_1$ and conduction band $c_3$ with temperature results in an infrared (IR) transition of up to 0.4 eV in the CDW phase. Excited electron-hole pairs are driven by our mid-IR field, generating harmonics. **c** Optical conductivity calculated from a Drude-Lorentz fit to the measurements in

ref. 44. As temperature decreases, a resonance emerges below $T_c$, corresponding to the $v_1$ to $c_3$ transition. **d** The reflectivity at the fundamental laser frequency (0.39 eV) and from ref. 44 for comparison. Linear fits find the transition temperature $197.3 \pm 3.5$ K. **e** Sketch of the reflection geometry in 45-degree incidence with θ the angle of polarization and α the crystal offset. The hexagon indicates 1st Brillouin zone orientation and the reduced Brillouin zone in the CDW phase.

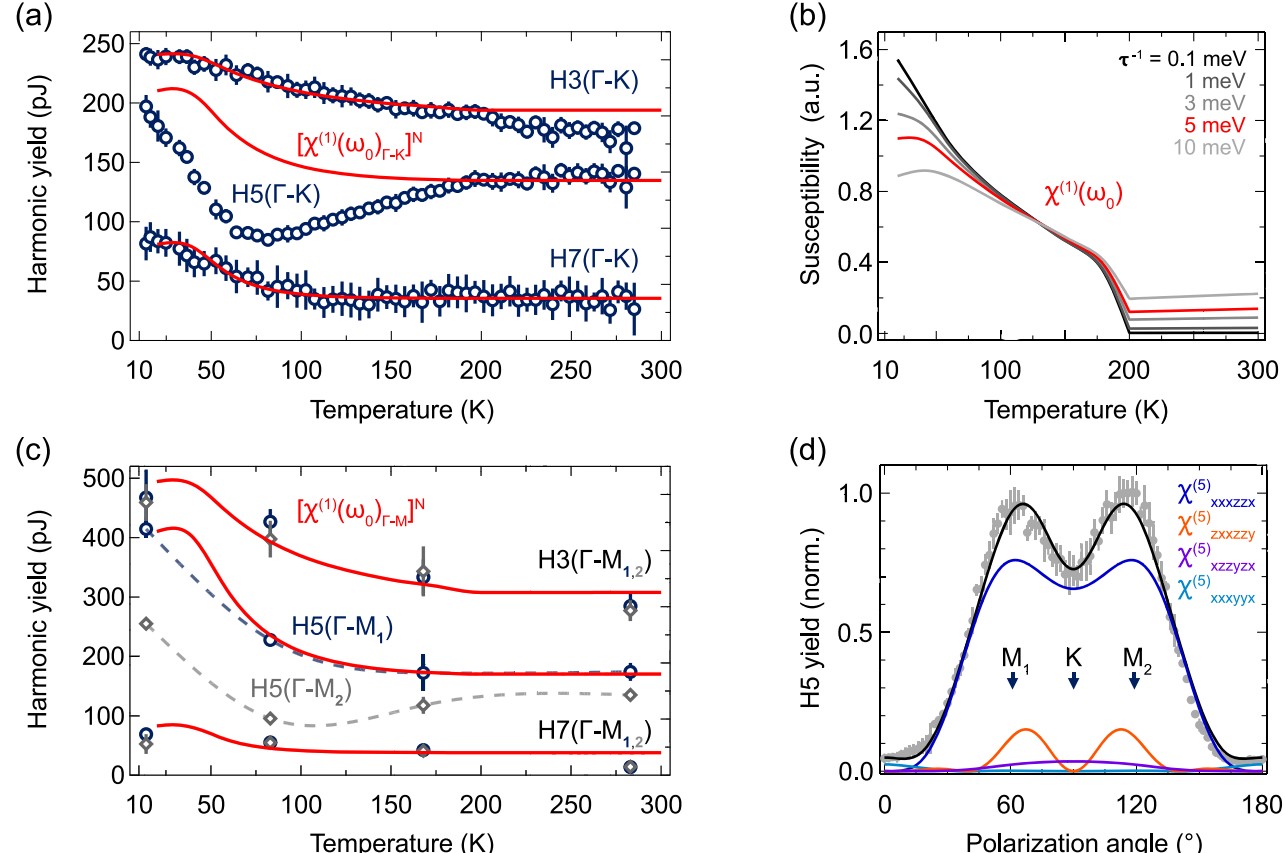

**Fig. 2 | Functional behavior of harmonics.** Harmonic yield for the 3rd, 5th, and 7th orders as a function of temperature. The scans were conducted with the driving field polarization along the (**a**) Γ-K direction and (**c**) Γ-M directions; square markers indicate the Γ-$M_2$ direction. We scale H5 and H7 by $10^4$ and $3 \times 10^5$ for visibility. Red lines represent the optical conductivity calculated in (**b**) from the mean-field model at the fundamental frequency and for a scattering rate of 5 meV to the respective power of the harmonic order. **d** Polarization angle scan for H5 at 283 K with fit based on $\chi^{(5)}_{ijklmn}$ tensor components. Error bars in all figures show one standard deviation.

symmetry-equivalent. Along Γ-$M_2$, H5 exhibits a similar minimum to that seen along Γ-K. In contrast, along Γ-$M_1$, H5 returns to the expected harmonic behavior and aligns well with predictions from Miller's rule.

A full angular scan of H5 in the high-temperature phase (Fig. 2d) reveals that the Γ-K direction forms a valley between two peaks along Γ-$M_1$ and Γ-$M_2$. This pattern results from the hexagonal lattice and the 45-degree incidence geometry, where harmonic yield is minimized for an s-polarized driving field. Within the perturbative framework, the angular dependence of H5 can be decomposed into contributions from the 5th-order nonlinear susceptibility tensor (see Methods). For $TiSe_2$, which has the point group P-3m1, there are 12 independent non-zero tensor components. Of these, only two—$\chi^{(5)}_{xxxzzx}$ and $\chi^{(5)}_{zxxzzy}$—produce the distinct double peak seen at Γ-$M_1$ and Γ-$M_2$, and dominate the fit used to match the measured data.

Additionally, four other components peak along Γ-K, helping to explain the contrast enhancement and the low-temperature minimum in that direction. However, below 80 K, the strong asymmetry in H5 between Γ-$M_1$ and Γ-$M_2$ cannot be captured by this model, as all tensor components are symmetric about 90° (see Supplementary Fig. 3).

Together, these two modeling approaches illustrate both the strengths and limitations of simple descriptions within the CDW phase. While the anomalous H5 anisotropy between Γ-$M_1$ and Γ-$M_2$ will be examined in the next section, we can already rule out a transition to non-perturbative dynamics as the cause. Furthermore, Miller's rule-based scaling still provides a reliable way to predict how harmonic yield changes with temperature—even within the CDW phase and for potentially non-perturbative harmonics like H7. Finally, combining harmonic measurements with our mean-field model allows us to extract scattering rates in the CDW phase without requiring full broadband spectral scans.

## High harmonic tomography of the CDW

To further investigate the stark differences between harmonic orders and to understand the specific sensitivity of H5 to the CDW phase-transition and the anomalous low temperature response, we conducted polarization scans across varying temperature regimes, controlling the input polarization with a half-wave plate.

Figure 3 displays the behavior of H3, H5, and H7 as functions of the fundamental polarization angle. An angle of 90° corresponds to the Γ-K direction, while 60 and 120° align with the Γ-$M_1$ and Γ-$M_2$ directions, respectively. Near room temperature and above $T_c$, both H3 and H7 exhibit a peak for p-polarized input, while H5 splits into two symmetric peaks at 60° intervals, thus registering the hexagonal crystal symmetry. This is consistent with Fig. 2d.

As the sample is cooled below the phase transition, all harmonic peaks increase in intensity, with the contrast between H5's double peaks also becoming more pronounced, as seen in Fig. 3d, marked by the diminishing minimum between them. Below 100 K, H5 develops an asymmetry, with the peak at 60 degrees rising sharply at lower temperatures. This asymmetry also emerges in H7, where the center of mass shifts to the right, while H3 exhibits a slight but detectable rotation in the opposite direction.

Comparing H5's temperature scaling with Fig. 2 for high-symmetry directions, the response along Γ-$M_1$ follows the mean-field behavior, but clear deviations occur along the Γ-K direction. Interestingly, the Γ-$M_2$ direction at 120°, which should be symmetry-equivalent, shows similar scaling to Γ-K.

## Discussion

These, at first glance, unexpected features reflect the microscopic changes occurring in the sample, which High Harmonic Spectroscopy detects sensitively. First, the overall increase in harmonic yield

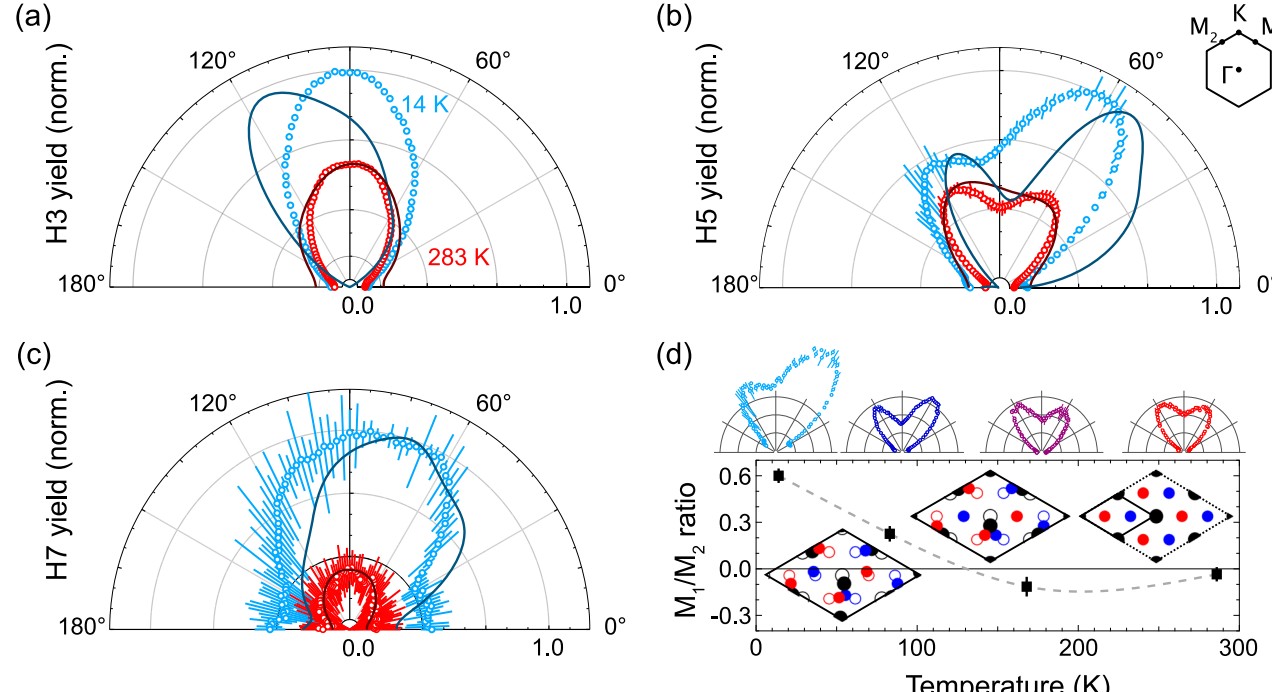

**Fig. 3 | Ellipsometry of harmonic emission.** Harmonic yields for H3 (**a**), H5 (**b**), and H7 (**c**) as a function of the driving field polarization at 283 K and 14 K. The sketch illustrates the corresponding Brillouin zone orientation, with 90° indicating p-polarization of the driving field along Γ-K. Solid lines represent the simulated angular dependence of the harmonics based on the mean-field model for the respective temperatures. **d** The relative ratio of H5 peaks at 60° and 120° as a function of temperature and trend line. Insets on top display the H5 angular distribution. The supercells indicate the atom positions for Ti (black), Se1 (red), and Se2 (blue) for no displacement (normal) and exaggerated equal or unequal displacement in the CDW. Error bars in all figures show one standard deviation.

at lower temperatures sensitively detects the phase transition, indicating that the ultrashort driving field can coexist with the CDW phase. This points to the robustness of the CDW phase against perturbations. Nevertheless, care was taken to limit the peak intensity and exposure to the sample, ensuring the experiment remained in the perturbative regime and moderate field amplitudes.

Notably, H5 exhibits a unique sensitivity to the crystal symmetry, differentiating between the zigzag (Γ-K) and armchair (Γ-M) directions. This can be attributed to the bandgap at the M point, which increases up to 1.95 eV (Fig. 1a), aligning closely with the H5 energy of 1.96 eV, creating a near-resonant transition. In contrast, the K point's bandgap is slightly lower at 1.83 eV, favoring a response to alignment along the Γ-M direction for H5. The increasing contrast below $T_c$, as shown in Fig. 3d, with the deepening Γ-K minimum and the enhanced Γ-M maximum, follows the renormalization of bands away from the Γ point.

Furthermore, the asymmetry between the theoretically equivalent Γ-$M_1$ and Γ-$M_2$ directions suggests an inequivalence within the CDW formation. A closer look at Fig. 3d reveals two distinct asymmetry features in the H5 double peak. The first, a subtle negative asymmetry (where $M_2$ exceeds $M_1$), is already present at room temperature and becomes more pronounced around 150 K. Below 100 K, a strong positive asymmetry ($M_1 > M_2$) dominates, even influencing the harmonic yield in Γ-K direction.

Turning to our mean-field model, which was used to calculate the harmonic response for different polarization angles (see Methods), we observe very good agreement in the high-temperature phase for all harmonics. The notable splitting of H5 along the crystal axes is accurately reproduced. To explore the origins of the asymmetry, we considered potential causes, such as a slight rotational misalignment of the crystal during mounting. XRD analysis confirmed a 7° ± 2° offset between the crystal axis and the p-polarization of the laser, which could indeed account for the negative asymmetry observed at 283 K, as simulated for a 5° offset.

At high temperatures, High Harmonic Spectroscopy can, therefore, determine crystal orientation and also identify the rotation direction.

However, the pronounced low-temperature asymmetry, which we further investigate in the tensor analysis cannot be explained with this approach and demands further investigation. We note that recent studies suggest that the CDW phase of TiSe₂ is chiral[19,37] at low temperatures and shows gyrotropic response[20]. Guided by this, we performed simulations to extract the response of the low-temperature phase to small ellipticity in the driving field and found only negligible differences compared with linear polarization. This rules out that the asymmetry is an artifact caused by the driving field.

Ultimately, only an asymmetry in the charge density wave strength $\Delta Q$ along the three Q directions could produce such a pronounced effect whilst preserving the CDW commensuration; see Fig. 3d. When the simulation assumed $\Delta_{Q_1} \neq \Delta_{Q_2} \neq \Delta_{Q_3}$, a stronger asymmetry in H5 emerged, consistent with the experimental observation at 14 K. This anisotropic CDW dominates over the much smaller offset effect seen at higher temperatures. Notably, the center of mass shifts of H5 and H7 to the right and H3 to the left are also reproduced, though slightly overestimated, supporting the idea of an inequivalence between atomic displacements in the TiSe₂ CDW phase below 100 K.

A possible onset of strain at these temperatures could create such a distortion. This is in accord with recent scanning tunneling microscopy measurements[38]. While this is largely a lattice effect, it is worth noting that the Γ-K direction, which shows a significant signal increase at 14 K, aligns with the atomic displacement direction and the relevant phonon modes. Another explanation may lie in the backfolding of bands and their orbital character[39]. While Fig. 1a illustrates the three conduction bands $c_1$, $c_2$, and $c_3$ along the Γ-M path, they split along the M directions, and only Γ-$M_3$ has a band nearly resonant with H5. This strongly suggests that our experiments sensitively detect the concomitant hybridization between Se p-orbitals and Ti $d_{xy}$, $d_{yz}$, or $d_{zx}$ orbitals.

## Conclusion

This work advances our insight into the transitions within the correlated charge density wave (CDW) phase of TiSe₂ by leveraging polarization-

resolved HHG spectroscopy. Distinct from prior research, which primarily focused on crystallographic changes, this study delves into the reordering occurring within the CDW phase as the material is cooled from room temperature to 14 K. By correlating ultrafast field-driven dynamics with the material's potential landscape, we demonstrate that HHG can coexist with the CDW and is exceptionally sensitive to symmetry breaking due to the phase transitions and lattice dynamics. Our findings unveil an anisotropic component far below the CDW transition temperature while probing the backfolding of specific bands, shedding light on the nature of this phase. This investigation underscores the complex interplay between linear and nonlinear optical responses and their deviation from the usual perturbative/ non-perturbative dichotomy, providing a novel perspective on the behavior of correlated quantum phases in condensed matter systems.

## Methods

### Experiment

The experiment was conducted using a home-built mid-infrared OPCPA laser system, delivering carrier-envelope phase (CEP) stable pulses of 100 fs duration at a repetition rate of 160 kHz and a central wavelength of 3.2 μm. By focusing the beam with a 150 mm lens, we achieved peak intensities of up to $81.6 \pm 13.7$ GW cm$^{-2}$, allowing us to observe harmonic orders H3, H5, and H7 in a reflection geometry with a 45° incidence angle. The TiSe$_2$ sample, exfoliated from a larger bulk material to ensure a smooth surface, was placed in the vacuum chamber of a He cryostat. A UV-FS lens was used to image the beam into a spectrometer (OceanOptics Maya). To enhance the dynamic range for higher-order harmonics, a KG3 filter was employed to suppress the fundamental wavelength and attenuate H3. Alternatively, the reflected fundamental was measured using a power meter (Thorlabs S401C). Scans were repeated up to 15 times and averaged. The theory accounted for the change of field amplitude in-plane and out-of-plane when rotating the polarization. Scans to measure harmonic scaling with the pulse energy are shown in Supplementary Fig. 2, showing close to perturbative scaling for H3 and H5 from power law fits, while H7 shows a turn over to non-perturbative scaling.

### Tensor component analysis

In harmonic spectroscopy the harmonic angle dependence is modeled with the nonlinear susceptibility tensors[36,40,41]. For H5, we write the non-linear polarizability as $P_i^{(5)} \propto \sum_{ijklmn} \chi_{ijklmn}^{(5)} E_i E_j E_k E_l E_m E_n$ with $\chi_{ijklmn}^{(5)}$ the 5th order nonlinear susceptibility tensor and $A_0 \left( \frac{\sqrt{2}}{2} \cos\theta, \; \sin\theta, \; \frac{\sqrt{2}}{2} \cos\theta \right)$ the electric field vector after accounting for projection for 45° incidence. For TiSe$_2$ having the point group #164 P-3m1, only 12 unique tensor components are non-zero. The exact form was obtained with TENSOR from the Bilbao Crystallographic Institute[42]. The harmonic yield is proportional to the susceptibility and electric field as $Yield_N \propto \varepsilon_0^2 |\chi^{(N)} E^N|^2$.

We identify two tensor components ($\chi_{xxxzzx}^{(5)}$ and $\chi_{xxxzzy}^{(5)}$) which produce a strict double peak feature similar to the experiment with 5 more affecting the contrast around 90° specifically; see Supplementary Fig. 3. More importantly, all components are symmetric around the field orientation of 90 degrees and therefore only the symmetric H5 signals are generated.

### Mean-field theory

We employ a minimal phenomenological mean-field model to describe the charge density wave (CDW) order in TiSe$_2$. The Hamiltonian is expressed as

$$H = \sum_{\mathbf{k}} \left[ \varepsilon_v(\mathbf{k}) - \mu \right] d_{v\mathbf{k}}^\dagger d_{v\mathbf{k}} + \left[ \varepsilon_c(\mathbf{k}) - \mu \right] d_{c\mathbf{k}}^\dagger d_{c\mathbf{k}} + \sum_{\mathbf{Q},\mathbf{k}} \Delta_{(-\mathbf{Q})} d_{v\mathbf{k}+\mathbf{Q}}^\dagger d_{c\mathbf{k}}$$
$$+ \Delta_{(+\mathbf{Q})} d_{c\mathbf{k}-\mathbf{Q}}^\dagger d_{v\mathbf{k}}$$

Here, $\varepsilon_{v,c}(\mathbf{k})$ represent the valence and conduction band dispersions of TiSe$_2$, obtained from a tight-binding model[39], and $d_{v,c\mathbf{k}} (d_{v,c\mathbf{k}}^\dagger)$ is the corresponding creation (annihilation) operator. The parameters for the tight-binding model can be found in Ref. 39. As shown in Fig. 1, the valence band, primarily composed of Se 4p orbitals, forms a hole pocket centered at the Γ

point, while the conduction band, dominated by Ti 3 d orbitals, creates three symmetry-equivalent electron pockets at the M points. The interaction leads to a triple-**Q** CDW order, which couples the valence band hole pocket to the conduction band electron pockets, with wave vectors $|\mathbf{Q}_1| = |\mathbf{Q}_2| = |\mathbf{Q}_3| = |\Gamma M|$. The CDW order parameter is temperature dependent and becomes nonzero below T$_c$ ≈ 200 K. As a result of the CDW order, the Brillouin zone folds. In the reduced Brillouin zone (RBZ), the mean-field Hamiltonian can be recast as

$$H = \sum_{\mathbf{k} \in \text{RBZ}} \Psi_{\mathbf{k}}^\dagger H(\mathbf{k}) \Psi_{\mathbf{k}},$$

where we have the basis $\Psi_{\mathbf{k}} = \left( d_{v\mathbf{k}}, d_{c\mathbf{k}}, d_{v\mathbf{k}+\mathbf{Q}_1}, d_{c\mathbf{k}+\mathbf{Q}_1}, d_{v\mathbf{k}+\mathbf{Q}_2}, d_{c\mathbf{k}+\mathbf{Q}_2}, d_{v\mathbf{k}+\mathbf{Q}_3}, d_{c\mathbf{k}+\mathbf{Q}_3} \right)^T$ and $H(\mathbf{k})$ is the $8 \times 8$ momentum-dependent Hamiltonian matrix. A schematic of the normal and reduced Brillouin zone is provided in Supplementary Fig. 1.

### Optical conductivity

The optical conductivity of the material is expressed as

$$\sigma_{ij}(\omega) = \frac{ie^2}{\omega V} \left( \sum_{\mathbf{k},mn} \frac{\left[ f_m(\mathbf{k}) - f_n(\mathbf{k}) \right] H_{i,mn}(\mathbf{k}) H_{j,nm}(\mathbf{k})}{i0^+ + \varepsilon_{\mathbf{k}m} - \varepsilon_{\mathbf{k}n}} \right.$$
$$\left. + \sum_{\mathbf{k},m} \left[ U^\dagger(\mathbf{k}) \partial_i \partial_j H(\mathbf{k}) U(\mathbf{k}) \right]_{mm} f_m(\mathbf{k}) \right)$$

$$- \frac{ie^2}{V} \sum_{\mathbf{k},mn} \frac{\left[ f_m(\mathbf{k}) - f_n(\mathbf{k}) \right] H_{i,mn}(\mathbf{k}) H_{j,nm}(\mathbf{k})}{\left( i0^+ + \varepsilon_{\mathbf{k}m} - \varepsilon_{\mathbf{k}n} \right) \left( \omega + i0^+ + \varepsilon_{\mathbf{k}m} - \varepsilon_{\mathbf{k}n} \right)}$$

where $H_{i,mn}(\mathbf{k}) \equiv \left[ U^\dagger(\mathbf{k}) \partial_i H(\mathbf{k}) U(\mathbf{k}) \right]_{mn}$ with $U(\mathbf{k})$ being the unitary transformation diagonalizing the Hamiltonian, and $f_m(\mathbf{k}) = 1/\left( 1 + e^{\varepsilon_{\mathbf{k}m}/k_B T} \right)$ is the Fermi-Dirac distribution with $\varepsilon_{\mathbf{k}m}$ being the $m$th eigenvalue of $H(\mathbf{k})$. Here $V$ is the crystal volume and $e$ is the charge unit. We note that the conductivity has both the interband ($m \neq n$) and intraband ($m = n$) contribution, and the diagonal part of $\left[ f_m(\mathbf{k}) - f_n(\mathbf{k}) \right] / \left( \varepsilon_{\mathbf{k}m} - \varepsilon_{\mathbf{k}n} \right)$ with $m = n$ must be interpreted as $\partial f(\varepsilon_{\mathbf{k}m}) / \partial \varepsilon_{\mathbf{k}m}$.

### High harmonic generation

We now calculate the high harmonic spectra of TiSe$_2$, which is a nonlinear optical process producing higher-order harmonics of the incident frequency when an intense laser field interacts with the material. We note that the coupling of a material to the laser field in dipole approximation amounts to the replacement $\mathbf{k} \rightarrow \mathbf{k}(t) = \mathbf{k} + \mathbf{A}(t)$ in the Bloch Hamiltonian using the Peierls substitution, where $\mathbf{A}(t) = \left( A_x(t), A_y(t) \right)$ with

$$A_x(t) = \left[ \sin\theta \sin(\pi/3 - \alpha) + \left( \sqrt{2}/2 \right) \cos\theta \cos(\pi/3 - \alpha) \right] A(t),$$

$$A_y(t) = \left[ \sin\theta \cos(\pi/3 - \alpha) - \left( \sqrt{2}/2 \right) \cos\theta \sin(\pi/3 - \alpha) \right] A(t),$$

and $A(t) = A_0 \sin^2 \left( \frac{\omega t}{2 n_{\text{cyc}}} \right) \sin(\omega t)$, is the vector potential of amplitude $A_0$, center frequency $\omega$, and cycle number $n_{\text{cyc}}$, projected onto the material plane (the laser field has an incident angle 45°). The dimensionless amplitude is defined as $A_0 = \frac{e E_0 a}{\hbar \omega}$, with the experimental field strength of $E_0 = 0.078 \pm 0.007$ V Å$^{-1}$ (outside the material). Here $\theta$ is the polarization angle and $\alpha$ is the crystal axis offset. In this work, we set $A_0 = 1.2$, $\omega = 0.4$ eV, and $n_{\text{cyc}} = 8$.

Using the velocity gauge equation of motion in the Bloch basis[43], the time-dependent Schrödinger equation for the density matrix $\rho(\mathbf{k}, t)$ at

momentum $\mathbf{k}$ can be written as

$$\frac{d}{dt}\rho(\mathbf{k}, t) = -i\left[H\left(\mathbf{k} + \mathbf{A}(t); \{\Delta_{\mathbf{Q}_i}(t)\}\right), \rho(\mathbf{k}, t)\right]$$

where $\Delta_{\mathbf{Q}_i}(t) = (U_{\mathbf{Q}_i}/V)\sum_{\mathbf{k}}\text{Tr}[\rho(\mathbf{k}, t)d^{\dagger}_{\nu\mathbf{k}+\mathbf{Q}_i}d_{c\mathbf{k}}]$ is the time-evolved CDW order parameter. Here, $U_{\mathbf{Q}_i}$ denotes the effective interaction strength between electrons, which is taken as a phenomenological temperature-dependent parameter in our modeling to obtain suitable strength of initial CDW order parameters self-consistently. The initial state for each momentum is given by the Fermi-Dirac distribution, i.e., $\rho(\mathbf{k}, 0) = 1/(e^{\beta H(\mathbf{k})} + 1)$. Moreover, we find it useful to introduce a phenomenological dephasing effect as follows. After each time step $\delta t$ of the evolution governed by the above equation, we transform the density matrix into an adiabatic basis obtained by diagonalizing the instantaneous Hamiltonian $H\left(\mathbf{k} + \mathbf{A}(t); \{\Delta_{\mathbf{Q}_i}(t)\}\right)$, i.e., $\widetilde{\rho}(\mathbf{k}, t) = U^{\dagger}(\mathbf{k}, t)\rho(\mathbf{k}, t)U(\mathbf{k}, t)$, apply the dephasing as $\widetilde{\rho}_{mn}(\mathbf{k}, t) \rightarrow \widetilde{\rho}_{mn}(\mathbf{k}, t)e^{-\delta t/\tau}$ for $m \neq n$ with $\tau$ being the dephasing time, and then transform back to the Bloch basis. In this work, we set $U_{\mathbf{Q}_1} = -1.80965$ eV, $U_{\mathbf{Q}_2} = -1.80935$ eV, $U_{\mathbf{Q}_3} = -1.80905$ eV, $\Delta_{\mathbf{Q}_1} = 0.1148$ eV, $\Delta_{\mathbf{Q}_2} = 0.1147$ eV, and $\Delta_{\mathbf{Q}_3} = 0.1146$ eV for $T = 14$ K to simulate the anisotropic CDW order parameters in the low-temperature region. The corresponding dephasing time is chosen as $\tau = 0.01$ eV$^{-1}$. On the other hand, we have $\Delta_{\mathbf{Q}_i} = 0$ in the high-temperature semimetal phase, for which $U_{\mathbf{Q}_i}$ and $\tau$ are irrelevant.

We calculate the expectation value of the velocity operator in direction $j$

$$v_j(\mathbf{k}, t) = \text{Tr}\left[\rho(\mathbf{k}, t)\partial_{k_j}H\left(\mathbf{k} + \mathbf{A}(t); \{\Delta_{\mathbf{Q}_i}(t)\}\right)\right]$$

Denoting the polarization direction of the laser field as $\mathbf{n}_A$ and $v_j(t) = (1/V)\sum_{\mathbf{k}}v_j(\mathbf{k}, t)$, then the high harmonic spectrum is given by the Fourier transform of velocity $P(\omega) = \omega^2\left|\text{FFT}\left[\mathbf{v}\cdot\mathbf{n}_A\right]\right|^2$.

## Data availability
All data supporting the study are available as source data and with the corresponding data processing scripts on https://github.com/jbiegert/ICFO-AUO-TiSe2-HHG. All parameters necessary to reproduce the calculations are given in the "Methods".

## Code availability
The codes supporting the results in this work are available from the corresponding author upon reasonable request.

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

## Acknowledgements

J.B. acknowledges financial support from the European Research Council for ERC Advanced Grant "TRANSFORMER" (788218), ERC Proof of Concept Grant "miniX" (840010), FET-OPEN "PETACom" (829153), FET-OPEN "OPTOlogic" (899794), FET-OPEN "TwistedNano" (101046424), MINECO for Plan Nacional PID2020–112664 GB-I00; QU-ATTO, 101168628; AGAUR for 2017 SGR 1639, MINECO for "Severo Ochoa" (CEX2019-000910-S), Fundació Cellex Barcelona, the CERCA Programme/Generalitat de Catalunya, and the Alexander von Humboldt Foundation for the Friedrich Wilhelm Bessel Prize. JB also acknowledges Lasers4EU which is funded by the European Union under HEU-GA 101131771. I.T. and J.B. acknowledge support from Marie Skłodowska-Curie ITN "smart-X" (860553). ICFO-QOT group acknowledges support from the European Research Council for AdG NOQIA; MCIN/AEI (PGC2018-0910.13039/501100011033, CEX2019-000910-S/10.13039/501100011033, Plan National FIDEUA PID2019-106901GB-I00, Plan National STAMEENA PID2022-139099NB, I00, project funded by MCIN/AEI/10.13039/501100011033 and by the "EU NextGenerationEU/PRTR" (PRTR-C17.I1), FPI); QUANTERA MAQS PCI2019-111828-2; QUANTERA DYNAMITE PCI2022-132919, QuantERA II Programme co-funded by EU Horizon 2020 program Grant No 101017733; Ministry for Digital Transformation and of Civil Service of the Spanish Government through the QUANTUM ENIA project call —Quantum Spain project, and by the European Union through the Recovery, Transformation and Resilience Plan - NextGenerationEU within the framework of the Digital Spain 2026 Agenda; Fundació Cellex; Fundació Mir-Puig; Generalitat de Catalunya (European Social Fund FEDER and CERCA program, AGAUR Grant No. 2021 SGR 01452, QuantumCAT\U16-011424, co-funded by ERDF Operational Program of Catalonia 2014-2020); Barcelona Supercomputing Center MareNostrum (FI-2023-3-0024); HORIZON-CL4-2022-QUANTUM-02-SGA PASQuanS2.1, 101113690, EU Horizon 2020 FET-OPEN OPTOlogic, Grant No 899794, QU-ATTO, 101168628, EU Horizon Europe Program (This project has received funding from the EU's Horizon Europe research and innovation program under grant agreement No 101080086 NeQSTGrant Agreement 101080086—NeQST); ICFO Internal "Quantum-Gaudi" project. U.B. is also grateful for the financial support of the IBM Quantum Researcher Program. T.G. acknowledges financial support from the Agencia Estatal de Investigación (AEI) through Proyectos de Generación de Conocimiento PID2022-142308NA-I00 (EXQUSMI), and that this work has been produced with the support of a 2023 Leonardo Grant for Researchers in Physics, BBVA Foundation. R.W.C. acknowledges support from the Polish National Science Centre (NCN) under the Maestro Grant No. DEC-2019/34/A/ST2/00081. E.C. and S.M.-V. acknowledge the financial support from the European Union (ERC AdG Mol-2D 788222, FET OPEN SINFONIA 964396), the Spanish MCIN (2D-HETEROS PID2020-117152RB-100, co-financed by FEDER, and Excellence Unit "María de Maeztu" CEX2019-000919-M, and 2DM PID2022-137078NB-100, co-financed by FEDER) and the Generalitat Valenciana (PROMETEO Program, PO FEDER Program IDIFEDER/2021/078. This study forms part of the Advanced Materials program and was supported by MCIN with funding from European Union NextGenerationEU (PRTR-C17.I1) and by Generalitat Valenciana. We thank Anna Palau and the technical staff from The Institute of Materials Science of Barcelona (ICMAB-CSIC) for assistance with the XRD measurements. We thank B. Rethfeld from RPTU Kaiserslautern for helpful discussions.

## Author contributions

J.B. conceived the project; I.T., L.V., and J.P. conducted the measurements with support from J.B.; S.M.-V. and E.C. produced the samples; U.B., R.C., T.G., L.Z., and M.L. developed the mean-field theory and L.Z. conducted simulations; I.T., L.V., and J.P. carried out the data analysis with support by J.B.; I.T. and J.B. wrote the manuscript with input from the authors.

## Competing interests

The authors declare no competing interests.
