## [Transparent Peer Review file · Communications Materials]

High harmonic spectroscopy reveals anisotropy of the Charge-Density-Wave phase transition in TiSe₂

Corresponding Author: Professor Jens Biegert

Version 0:

Decision Letter:

Dear Professor Biegert,

Thank you for submitting your manuscript, "High harmonic spectroscopy reveals anisotropy of the Charge-Density-Wave phase transition in TiSe₂", to Communications Materials. It has now been seen by 2 referees, whose comments are appended below. You will see that while they find your work of potential interest, they have raised substantial concerns that must be addressed. In light of these comments, we cannot accept the manuscript for publication, but are interested in considering a revised version that addresses these serious concerns, in particular those concerning the interpretation of your results.

We hope you will find the referees' comments useful as you decide how to proceed. Should further experimental data or analysis allow you to address these criticisms, we would be happy to look at a substantially revised manuscript. However, please bear in mind that we will be reluctant to approach the referees again in the absence of major revisions. If the revision process takes significantly longer than three months, we will be happy to reconsider your paper at a later date, as long as nothing similar has been accepted for publication at Communications Materials or published elsewhere in the meantime.

When submitting your revised manuscript, please include the following:

-A response letter with a point-by-point reply to each of the referee comments and a description of changes made. Please include the complete referee report in the response letter. Please note that the response letter must be separate to the cover letter to the editors.

-A marked-up version of the manuscript with all changes to the text in a different colored font. Please do not include tracked changes or comments. Please select the file type 'Revised Manuscript - Marked Up' when uploading the manuscript file to our online system.

-A clean version of the manuscript. Please select the file type 'Article File'.

-An updated <https://www.nature.com/documents/nr-editorial-policy-checklist.zip> Editorial Policy checklist, uploaded as a 'Related Manuscript File' type. This checklist is to ensure your paper complies with all relevant editorial policies. If needed, please revise your manuscript in response to these points. Please note that this form is a dynamic 'smart pdf' and must therefore be downloaded and completed in Adobe Reader. Clicking this link will download a zip file containing the pdf.

Please use the following link to submit your revised manuscript files:

Link Redacted

** This url links to your confidential home page and associated information about manuscripts you may have submitted or be

reviewing for us. If you wish to forward this email to co-authors, please delete the link to your homepage first **

Please do not hesitate to contact me if you have any questions or would like to discuss the required revisions further. Thank you for the opportunity to review your work.

Best regards,

Klaas-Jan Tielrooij, PhD
Editorial Board Member
Communications Materials
orcid.org/0000-0002-0055-6231

Reviewers' comments:

Reviewer #1 (Remarks to the Author):

In this manuscript, Tyulnev and co-authors have utilized higher harmonic generation (HHG) to study the charge density state in a transition metal dichalcogenide (TMD) 1T-TiSe₂. This TMD undergoes a 2x2x2 periodic lattice distortion near ~200K, which is believed to be a signature of exciton condensation in the material. The authors excited the sample with 3.2 μm pulsed MIR laser and measured the reflectivity at the fundamental (H1) as well as 3rd (H3), 5th (H5), and 7th (H7) harmonic frequencies. All of these responses increased monotonically below T_c aside from H5. Furthermore, by varying the incident polarization, the authors observed a pronounced asymmetry in the H5 response along symmetry-equivalent crystallographic directions. The authors interpret these observations as evidence for the breakdown of the symmetric 2x2x2 modulation and as a demonstration of the HHG as a sensitive probe of CDWs.

While HHG is an emerging experimental probe, highly sensitive to important material parameters including symmetry, band structure, Berry phase effects, and so on, I am not convinced that the current experiment adds any useful insight on what is known about the CDW phase in 1T-TiSe₂ or excitonic insulators in general. Furthermore, the conclusions drawn here appear to be highly speculative and stand in sharp contrast to almost every other experiment on this well-studied TMD.

Let me highlight a few of these conflicts and inconsistencies:

1. In Fig. 1d, the authors fit the reflectivity at 3.2μm to the BCS gap equation and argue that it can be used as a proxy for order parameter with rather insubstantial justification in the following section. While optical conductivity can be used to determine the BCS order parameter, one typically deploys a sum-rule analysis to do so. Below T_c, there is a depletion of spectral weight in the real part of optical conductivity and the integral of the depletion region is related to the superfluid density by the so-called Ferrell-Glover-Tinkham sum rule, which in turn is related to the order parameter. Alternatively, one can fit the spectra over large frequency range (using Mattis-Bardeen formula or some variant thereof) to extract the gap, Δ. However, due to the spectral weight rearrangement, the conductivity (and by extension other optical constants) in the vicinity of 2Δ is always modified as one approaches T_c. Therefore, while optical constants at a frequency ω~2Δ is likely to be temperature independent at T>T_c (if ω>kBT), they can become temperature dependent near and below T_c [Basov et.al. RMP 83(2), 471-541 (2011), Dressel Adv. in CMP, 2013(1), 104379]. Nevertheless, this DOES NOT imply that that reflectivity or conductivity at a single frequency close to 2Δ can be a proxy for the BCS order parameter, without rigorous calculations involving realistic material parameters to model the response [Chen PRL 71(14), 2304 (1993)]. The trend in Fig. 1d probably fits just as well to a straight line with a finite slope below T_c.
2. The authors have strongly criticized the protocol for gap measurements in ARPES, and while I agree that it can indeed be tricky in some instances, especially where bands are not well resolved, this particular TMD is an exception. Not only there has been multiple high-quality synchrotron-based measurements over the years, TiSe₂ has also been studied with time-resolved ARPES by multiple groups which mapped out both the valence and conduction bands. Moreover, the estimates for the gap from other probes such as STM and optics are also remarkably consistent with one another. Therefore, this is probably not a pertinent criticism in this context.
3. In line #117-199, the authors propose Miller's rule type estimate of susceptibility, but it is not clear to me if this is being used later or how it is relevant to the rest of the discussion. It is not even clear if such simplistic assumption is sensible in the present context. In the following paragraph, the authors propose a power law in reduced temperature to model the temperature dependence of the HHG and seemingly an exponent approximately equal to the (harmonic order)/2 works well for H3 and H7. Subsequently, the authors write, "This scaling matches the perturbative description of the nonlinear susceptibility" but this is perturbative in temperature and not electric field. Why does the nonlinearity depend on the reduced temperature in this fashion?
4. The interpretation of the polarization analysis is perhaps the most problematic aspect of the paper. The asymmetry in the CDW wavevector has never been observed in any experiment as far as I am aware, and the authors do not cite any reports to that effect. While there is some disagreement regarding the origin of the periodic modulation and such, every momentum-resolved measurement, be it x-rays, neutron, STM or ARPES, have found a 2x2x2 modulation and it is consistent with lattice dynamics studies through Raman and IR spectroscopies as well. Multiple theories have even explained this quantitatively and a summary of these results can be found in Ref. 40 for instance. It is therefore extremely unlikely that a deviation from the 2x2x2 distortion would be missed in all of these measurements. Furthermore, because the 2x2x2 superstructure is a

commensurate CDW, a transition to unequal wavevector would involve a commensurate-incommensurate transition, which must be first order [McMillan PRB, 12(4), 1187 (1975)]. If such a transition exists between 80-100K as the authors posit, a hysteretic anomaly must appear in transport measurements. None of these contradictions are addressed in the paper, and in light of the overwhelming lack of supporting evidence, I find the proposed hypothesis highly implausible.

5. Strain could be important here and it is certainly possible that the samples are getting strained at low temperatures due to some uncompensated thermal contractions. It is known that the CDW in TMDs and in Tise2 becomes incommensurate under pressure/strain/intercalation etc. However, in such a scenario, the authors should investigate quantitatively why H5 gets modulated while H3 and H7 appear to be less sensitive. Strong optical field itself can also strain the material, especially through piezo-optical coupling, but a comprehensive symmetry analysis is necessary to assess the feasibility of such process.

6. More importantly perhaps, it must be noted that similar asymmetry between HHG response for polarizations along symmetry-equivalent directions have been observed before in TMDs [Yoshikawa et. al. Nat. Comm. 10, 3709 (2019)]. These measurements also found differences between zigzag and armchair directions, and all of these effects could be explained by band structure related effects without invoking strong correlation physics, CDW or otherwise. The observations reported here could likely be a manifestation of similar phenomena.

7. The authors allude to quantum phase transitions several times (abstract, introduction, conclusion) and goes on to write: "we demonstrate that HHG is exceptionally sensitive to symmetry breaking due to the quantum phase transitions and lattice dynamics". However, a transition at 200K (or 80K) is most certainly not a quantum phase transition. Which transition are the authors alluding to?

8. The data along G-M direction has 3-4 points. Fitting it to a non-linear function with two (or more) parameters is highly questionable.

Overall, I do not find the results to be novel. Systematic study of HHG (up to 18th order) has been demonstrated in multiple TMDs over 5 years ago. Many aspects of the present dataset were also observed in the previous work including the non-monotonic harmonic intensities and polarization sensitivity. While the data might be technically sound, the authors fail to provide reliable evidence for their inferences. I do not recommend this manuscript for publication.

Reviewer #2 (Remarks to the Author):

The paper "High harmonic spectroscopy reveals anisotropy of the Charge-Density-Wave phase transition in TiSe2 " by I. Tyulnev et al. presents an investigation on polarization-resolved high-order harmonic generation spectroscopy (HHGS) in TiSe2 as a function of the temperature through a correlated charge density wave (CDW) phase transition.

The major outcome of this study is the demonstration of anisotropic 5th harmonic generation emission as a function of polarization in the charge-density-wave (CDW) phase. This finding could indicate band splitting along different M-directions in the CDW phase, a phenomenon that harmonic spectroscopy can probe with exceptional sensitivity.

The experimental results presented in the paper are of high quality. However, the discussion and interpretation could be improved to provide a clearer and more comprehensive presentation of the findings.

Harmonic spectroscopy is a novel approach, and its theoretical interpretation often relies on models where certain parameters are empirically adjusted to match experimental observations. This appears to be the case in this paper, although it is not explicitly stated. That said, high-quality experimental results like those presented here should not be undervalued simply due to the lack of direct methods for reproducing the findings or the absence of a definitive theoretical explanation. The intriguing physics observed through the experiment is compelling. Moreover, additional measurements have been conducted to rule out possible misalignments of the laser, further supporting the validity of the results.

Given the novelty and importance of these experimental findings, particularly within the framework of this emerging spectroscopic approach, I believe the work is well-suited for publication in Communications Materials. However, the authors should first address the following concerns regarding the presentation and interpretation of the results, as well as clarify a few specific points.

These are in particular the parts that I suggest the authors revise:

1. The manuscript states that "the experiment remained in the perturbative regime and moderate field amplitudes," supported by a perturbative model that reproduces the temperature dependence of harmonics 3 and 7. However, when discussing harmonic 5 and in the conclusions, the authors mention a "deviation from simple perturbative dynamics." Are we in the perturbative regime or not?

1a. A clear indication of a non-perturbative regime for harmonic 5 could have been provided by examining the harmonic intensity as a function of the driving field intensity. Was such an analysis attempted?

1b. Perturbative dynamics imply that the highly non-linear regime characteristic of high-harmonic generation (HHG) is not

accessed. Is this the case in this experiment? How many harmonics were observed (i.e., what is the cutoff)? Do the harmonics exhibit a plateau, or do they follow a power-law decay, characteristic of the perturbative regime?

1c. The deviation of harmonic 5 from harmonics 3 and 7 might not necessarily indicate non-perturbative behaviour. Instead, it could result from the “non-parametric” nature of harmonic 5. While the polarization may still be expressed as a Taylor expansion in susceptibility coefficients, these coefficients are sensitive to resonances at specific frequencies and crystal-axis orientations. The discussion before the conclusion suggests that a resonance overlapping harmonic 5 could render the susceptibility coefficient complex, leading to a highly non-parametric harmonic generation. Can the authors clarify if this is the correct interpretation of their experimental findings?

1. If it is confirmed that the experiment presents results of HHG in the perturbative regime, this should be clearly stated in the title, as HHG spectroscopy usually refers to the spectroscopy approaches that can be accessed from the non-perturbative description of the process. If the authors claim to be sensitive to highly non-linear processes, a better experimental evidence of this claim or support by theory should be provided.

2. The manuscript claims “excellent agreement” between the mean-field model and experimental results. However, in the methods section, the interaction strength and order parameter are assigned different values along the translation vectors Q . Were these values obtained by fitting to the experimental data? If so, this agreement stems from fitting rather than modeling, and this distinction should be explicitly stated.

3. A possible chirality in the electronic states would be significantly coupled to the lattice [See PRB 3 L022003 (2021)]. The observed asymmetry between translation vectors Q introduced in the model reproduces the experimental results. Does this asymmetry correspond to a chiral point group in the crystal lattice? If so, chirality cannot be entirely ruled out as a source of the observed asymmetries.

4. Before the conclusions, the authors say “Another explanation may lie in the backfolding of bands and their orbital character. While Fig. 1a illustrates the three conduction bands c_1 , c_2 , and c_3 along the Γ -M path, they split along the M directions, and only Γ -M3 has a band nearly resonant with H5”. It would be useful to show this split along the M directions. What M direction is shown in Fig.1(a)? What is the Γ -M3 direction?

5. The manuscript states a peak intensity of 40 GW/cm². How was this value estimated? Does it refer to the field outside the sample or the transmitted field within it? Furthermore, simulations were performed at $A=0.12$ a.u., while the experimental field corresponds to $A=0.1$ a.u.. Why was this discrepancy introduced?

6. Rotating the polarization, instead of the sample, means that for different polarization angles, one must consider that s and p components behave differently at the sample/vacuum interface due to different Fresnel transmission coefficients for the driving field and reflection coefficients for the harmonics. This effect is especially relevant at a 45° incidence angle and could explain the near-zero HHG at 0° and 180°, where s-polarized driving fields might experience suppressed transmission. Polarization-resolved HHG is better performed when the sample is rotated instead of the driving field. I understand this may be impractical when using a cryostat for temperature-dependent measurements. However, this limitation, the related consequences, and the experimental signatures related to this geometry should be better explained. Moreover, it is not clearly stated if the model is taking into account this particular geometry.

7. The experimental results are presented only for angles ranging from 0° to 180°. Why was a full 360° scan not performed?

Communications Materials is committed to improving transparency in authorship. As part of our efforts in this direction, we are now requesting that all authors identified as ‘corresponding author’ create and link their Open Researcher and Contributor Identifier (ORCID) with their account on the Manuscript Tracking System prior to acceptance. ORCID helps the scientific community achieve unambiguous attribution of all scholarly contributions. You can create and link your ORCID from the home page of the Manuscript Tracking System by clicking on ‘Modify my Springer Nature account’ and following the instructions in the link below. Please also inform all co-authors that they can add their ORCIDs to their accounts and that they must do so prior to acceptance.

Version 1:

Decision Letter:

Dear Professor Biegert,

Thank you for submitting your revised manuscript, "High harmonic spectroscopy reveals anisotropy of the Charge-Density-Wave phase transition in TiSe₂", to Communications Materials. It has now been seen again by the 2 referees, whose comments are appended below. You will see that while they find your work improved, some points are still requiring minor revisions.

We remain interested in the possibility of publishing your study in Communications Materials, but would like to consider your response to these concerns in the form of a revised manuscript before we make a decision on publication.

We therefore invite you to revise and resubmit your manuscript, taking into account the points raised.

When submitting your revised manuscript, please include the following:

-A response letter with a point-by-point reply to each of the referee comments and a description of changes made. Please include the complete referee report in the response letter. Please note that the response letter must be separate to the cover letter to the editors.

-A marked-up version of the manuscript with all changes to the text in a different colored font. Please do not include tracked changes or comments. Please select the file type 'Revised Manuscript - Marked Up' when uploading the manuscript file to our online system.

-A clean version of the manuscript. Please select the file type 'Article File'.

-An updated <https://www.nature.com/documents/nr-editorial-policy-checklist.zip> Editorial Policy checklist, uploaded as a 'Related Manuscript File' type. This checklist is to ensure your paper complies with all relevant editorial policies. If needed, please revise your manuscript in response to these points. Please note that this form is a dynamic 'smart pdf' and must therefore be downloaded and completed in Adobe Reader. Clicking this link will download a zip file containing the pdf.

In the event that your manuscript is accepted we will provide detailed guidance on our journal policies and formatting. You may however wish to ensure that the manuscript complies with our house style at this stage. See our style and formatting guide (<https://www.nature.com/documents/commsj-phys-style-formatting-guide-accept.pdf>) and checklist (<https://www.nature.com/documents/commsj-phys-style-formatting-checklist-article.pdf>) for reference.

Data availability statements and data citations policy: All Communications Materials manuscripts must include a section titled "Data Availability" at the end of the Methods section or main text (if no Methods). More information on this policy, and a list of examples, is available at <http://www.nature.com/authors/policies/data/data-availability-statements-data-citations.pdf>.

- Accession codes for deposited data
- Other unique identifiers (such as DOIs and hyperlinks for any other datasets)
- At a minimum, a statement confirming that all relevant data are available from the authors
- If applicable, a statement regarding data available with restrictions
- If a dataset has a Digital Object Identifier (DOI) as its unique identifier, we strongly encourage including this in the Reference list and citing the dataset in the Data Availability Statement.

DATA SOURCES: We strongly encourage authors to deposit all new data associated with the paper in a persistent repository where they can be freely and enduringly accessed. We recommend submitting the data to discipline-specific, community-recognized repositories, where possible and a list of recommended repositories is provided at <http://www.nature.com/sdata/policies/repositories>.

If a community resource is unavailable, data can be submitted to generalist repositories such as <https://figshare.com/> figshare or <http://datadryad.org/> Dryad Digital Repository. Please provide a unique identifier for the data (for example a DOI or a permanent URL) in the data availability statement, if possible. If the repository does not provide identifiers, we encourage authors to supply the search terms that will return the data. For data that have been obtained from publically available sources, please provide a URL and the specific data product name in the data availability statement. Data with a DOI should be further cited in the methods reference section.

Please use the following link to submit your documents:

Link Redacted

We hope to receive your revised paper within three months; please let us know if you aren't able to submit it within this time so that we can discuss how best to proceed. If we don't hear from you, and the revision process takes significantly longer, we will close your file. In this event, we will still be happy to reconsider your paper at a later date, as long as nothing similar has been accepted for publication at Communications Materials or published elsewhere in the meantime.

Please do not hesitate to contact me if you have any questions or would like to discuss these revisions further. We look forward to seeing the revised manuscript and thank you for the opportunity to review your work.

Best regards,

Klaas-Jan Tielrooij, PhD
Editorial Board Member
Communications Materials
orcid.org/0000-0002-0055-6231

Reviewers' comments:

Reviewer #1 (Remarks to the Author):

I would like to thank the authors for their reply. The newly added references, calculations and revisions in the resubmitted manuscript certainly clarifies a number of important points which I found to be rather confusing in the previous draft.

The calculations of the harmonic yield based on the linear susceptibility and corresponding Miller's rule estimates are more concrete compared to the previous iteration. It is interesting to see the importance of the scattering rate in this regard, although it is perhaps not surprising. The point group symmetry analysis and revisions to Fig. 2 certainly make for a more compelling case for the anomalies associated with the 5th harmonics response.

Few general remarks:

1. Ref. 36 in the main text is probably incorrect. Based on Fig. 1 caption I was expecting to find the PRL (2007) from Li et. al. but the current manuscript on my end is referring to Boyd's nonlinear optics textbook instead.
2. It might be useful to add the Drude-Lorentz parameters to the supplemental. Temperature dependence of the scattering rate for the 0.4eV Lorentzian is certainly useful.

I am satisfied with the revised manuscript, and I would recommend the manuscript for publication.

Reviewer #2 (Remarks to the Author):

The authors have addressed most of the points raised in my previous review. I have only a couple of comments regarding the answers to my questions.

- In answering comment 4, the authors say: "We would like to keep the focus on the fact that the harmonics are probing the v1 and c3 bands, which are mainly participating within the three-step-model-like picture for HHG in solids and their gap at the M points being nearly resonant to H5." But then, in the new version of the paper, the authors now state: "Previous broadband studies (Fig. 1c) show that the optical response at harmonic frequencies is nearly independent of temperature. This suggests that harmonic generation is primarily influenced by changes at the fundamental frequency, where the resonance appears." This contradicts the previous statement that, being H5 close to a resonance, we must be highly sensitive to H5. Being $\chi(5\omega_0)$ at resonance, harmonic generation can be primarily influenced also by changes of the χ at the harmonic frequency (not only the fundamental). Fig. 1c only demonstrates that it is not dependent on the change in the conductivity. Please clarify.

- Comment 5. I thank the authors for the clarification and the amendments. I do not agree that $A_0 = 0.69$ and $A_0 = 1.2$ are equivalent for HHG. A change of almost two times in the field strength dramatically changes the result of simulations for HHG, even in the perturbative regime. I ask again to clarify in the paper why, in the simulation, a value close to the one measured was not used. What would be the result of the simulation by using this value?

- Check references in the new version of the manuscript, some have not been updated. I've found one in the methods

section "The parameters for the tight-binding model can be found in Ref. 40" but now Ref. 40 is not anymore the correct reference.

Final remarks:

I will start from one of the most important points raised by the first referee to elaborate my final assessment. The referee comments about the similar asymmetry between HHG response observed in TMDs [Yoshikawa et al., Nat. Commun. 10, 3709 (2019)]. I would like to add to the discussion that Yoshikawa et al. point to a different type of asymmetry (zigzag horizontal vs. armchair vertical). No change is observed or claimed between the horizontal and 60° directions, which are (and must be) completely equivalent directions in the absence of a change in the structure.

The new paragraph introduced by the authors, which includes a fit with the susceptibility tensor parameters, nicely shows that. The symmetry of TiSe₂ in the standard phase enforces the M1 and M2 directions to be equivalent. The high sensitivity of HHG to the structure and the reported HHG response as a function of the temperature strongly point to the fact that the effect observed is associated with the formation of the superlattice.

I, however, agree with the first referee that whether the specific effect observed is due to the backfolding of the bands in the low-temperature phase, the anisotropy of the CDW, or other sources of symmetry breaking associated with the phase transition, is still a matter of discussion. But in any case, I believe the authors have provided enough evidence that it must be related to the CDW phase.

That said, the theoretical results presented in the paper give sufficient indication that the CDW anisotropy can show up in HHG with quite a strong signature, and the good agreement with the experiment (even if some parameters are arbitrarily chosen) points to the fact that the explanation provided by the authors is reasonable.

I think these results merit publication in this journal.

I confirm my previous evaluation, and I believe the work is well-suited for publication in Communications Materials after addressing the minor comments above.

Communications Materials is committed to improving transparency in authorship. As part of our efforts in this direction, we are now requesting that all authors identified as 'corresponding author' create and link their Open Researcher and Contributor Identifier (ORCID) with their account on the Manuscript Tracking System prior to acceptance. ORCID helps the scientific community achieve unambiguous attribution of all scholarly contributions. You can create and link your ORCID from the home page of the Manuscript Tracking System by clicking on 'Modify my Springer Nature account' and following the instructions in the link below. Please also inform all co-authors that they can add their ORCIDs to their accounts and that they must do so prior to acceptance.

If you experience problems in linking your ORCID, please contact the Platform Support Helpdesk.

Version 2:

Decision Letter:

Dear Professor Biegert,

Your manuscript titled "High harmonic spectroscopy reveals anisotropy of the Charge-Density-Wave phase transition in TiSe₂" has now been seen again by our referees, whose comments appear below. In light of their advice I am delighted to say that we are happy, in principle, to publish a suitably revised version in Communications Materials.

We therefore invite you to edit your manuscript to comply with our journal policies and formatting style in order to maximise the accessibility and therefore the impact of your work.

EDITORIAL REQUESTS

* Your manuscript should comply with our policies and format requirements, detailed in our style and formatting guide (<https://www.nature.com/documents/commsj-phys-style-formatting-guide-accept.pdf>).

* Please edit your manuscript according to the editorial requests in the attached table, and outline revisions made in the right hand column. If you have any questions or concerns about any of our requests, please do not hesitate to contact me. It is important that each request be addressed in order to avoid delays in accepting your manuscript. Please upload the completed table with your manuscript files as a Related Manuscript file.

* The editorial requests table also includes a full list of the files that must be provided upon resubmission. Please upload your files according to this table.

* An updated editorial policy checklist that verifies compliance with all required editorial policies must be completed and uploaded with the revised manuscript. All points on the policy checklist must be addressed; if needed, please revise your manuscript in response to these points. Please note that this form is a dynamic 'smart pdf' and must therefore be downloaded and completed in Adobe Reader. Clicking this link will download a zip file containing the pdf.

OPEN ACCESS

Communications Materials is a fully open access journal. Articles are made freely accessible on publication. For further information about article processing charges, open access funding, and advice and support from Nature Research, please visit <https://www.nature.com/commsmat/open-access>

Please use the following link to submit your revised files:

Link Redacted

We hope to hear from you within two weeks; please let us know if the process may take longer.

Best regards,

Klaas-Jan Tielrooij, PhD
Editorial Board Member
Communications Materials
orcid.org/0000-0002-0055-6231

REVIEWERS' COMMENTS:

Reviewer #2 (Remarks to the Author):

The authors have thoroughly addressed all of my comments. I recommend the manuscript for publication in Communications Materials.

Subject Decision on manuscript COMMSMAT-24-0690

We thank the reviewers for carefully considering our work and appreciate their valuable feedback. We have carefully considered and addressed their comments in a detailed point-by-point response below.

Reviewer #1:

R1: While HHG is an emerging experimental probe, highly sensitive to important material parameters including symmetry, band structure, Berry phase effects, and so on, I am not convinced that the current experiment adds any useful insight on what is known about the CDW phase in 1T-TiSe₂ or excitonic insulators in general. Furthermore, the conclusions drawn here appear to be highly speculative and stand in sharp contrast to almost every other experiment on this well-studied TMD. Let me highlight a few of these conflicts and inconsistencies:

1. In Fig. 1d, the authors fit the reflectivity at 3.2 μ m to the BCS gap equation and argue that it can be used as a proxy for order parameter with rather insubstantial justification in the following section. While optical conductivity can be used to determine the BCS order parameter, one typically deploys a sum-rule analysis to do so. Below T_c , there is a depletion of spectral weight in the real part of optical conductivity and the integral of the depletion region is related to the superfluid density by the so-called Ferrell-Glover-Tinkham sum rule, which in turn is related to the order parameter. Alternatively, one can fit the spectra over large frequency range (using Mattis-Bardeen formula or some variant thereof) to extract the gap, Δ . However, due to the spectral weight rearrangement, the conductivity (and by extension other optical constants) in the vicinity of 2Δ is always modified as one approaches T_c . Therefore, while optical constants at a frequency $\omega \sim 2\Delta$ is likely to be temperature independent at $T \gg T_c$ (if $\omega \gg k_B T$), they can become temperature dependent near and below T_c [Basov et.al. RMP 83(2), 471-541 (2011), Dressel Adv. in CMP, 2013(1), 104379]. Nevertheless, this DOES NOT imply that that reflectivity or conductivity at a single frequency close to 2Δ can be a proxy for the BCS order parameter, without rigorous calculations involving realistic material parameters to model the response [Chen PRL 71(14), 2304 (1993)]. The trend in Fig. 1d probably fits just as well to a straight line with a finite slope below T_c .

We are grateful for the reviewer's assessment of our work and feedback. From the reviewer's points, we understand that the differences between the known and the unknown were not sufficiently clear, and this leads to an erroneous interpretation of what we present. For instance, the reviewer correctly states that a sum rule should be applied in a specific framework. The sum rule has to be used for the broadband infrared analysis due to the two band gaps arising from the three relevant bands. Our optical wavelengths do, however, only access the larger gap; thus, no such sum rule is required. We want to expand on this point based on the suggestion by the reviewer to make our point clear:

In Fig. R1, we show the energy-dependent optical conductivity for a variety of temperatures in TiSe₂. This plot was generated by fitting the reflectivity measurements from Ref. 1 with a Drude-Lorentz oscillator model. From this, we get all optical contributions over a wide energy range, i.e., phonon peaks, free-carrier response (Drude), and interband transitions (Lorentz).

Figure R1: Optical conductivity obtained from Drude-Lorentz fits to reflectivity data from G.Li. et al.¹ The interband v_1 to c_3 transition appears below 200 K centering at 0.4 eV.

Extracting the 2Δ gap is not trivial as it overlaps with the Drude response. As pointed out above, It can be estimated similarly, analogous to superconductors². For TiSe_2 , however, there is more than one gap due to the three conduction bands. The 2Δ gap corresponds only to the v_1 to $c_{1,2}$ transition corresponding to a spacing of $\Delta = 100$ meV at very low temperatures. Extracting this value would require, e.g., a sum-rule type analysis.

On the other hand, the relevant infrared transition, not to be confused with the broad infrared band as found in the traditional picture for superconductors, the v_1 to c_3 transition is around 400 meV and sufficiently far away from the Drude term and low-frequency responses. This feature is directly fitted with one Lorentz oscillator, whose resonance frequency directly indicates the gap energy. As the band shifts follow the order parameter, this shift should also follow it.

However, as reflectivity is calculated from a ratio between the optical responses, it is not a given that the reflectivity will have the same shape as the order parameter. Meanwhile, the temperature-dependent oscillator strength and scattering rate will further modify this shape.

Figure R2: Reflectivity of TiSe_2 as a function of temperature. Approximately linear scaling of the reflectivity close to T_c allows determination of the transition temperature $T_c = 197 \text{ K} \pm 3.5 \text{ K}$.

The referee correctly points out that the reflectivity is approximately linear close to the transition temperature T_c . With the objective of the fit being to extract the transition temperature from the reflectivity measurement, we, therefore, employ linear fits in Fig. R2 and get a similar T_c (within the error) as previously reported, now at $197 \text{ K} \pm 3.5 \text{ K}$. We hope this clarifies the point.

2. The authors have strongly criticized the protocol for gap measurements in ARPES, and while I agree that it can indeed be tricky in some instances, especially where bands are not well resolved, this particular TMD is an exception. Not only there has been multiple high-quality synchrotron-based measurements over the years, TiSe₂ has also been studied with time-resolved ARPES by multiple groups which mapped out both the valence and conduction bands. Moreover, the estimates for the gap from other probes such as STM and optics are also remarkably consistent with one another. Therefore, this is probably not a pertinent criticism in this context.

We did not intend to give the impression of discarding the large body of excellent prior work on the subject; instead, we wanted to draw attention to differences. In our article, we followed the excellent discussion and conclusions by C. Monney³ about their ARPES measurements, compared to the more recent RIXS measurements where direct observation of the c_3 band was achieved. While c_1 and c_2 bands are reachable via ARPES, direct observation of c_3 remains mainly a feature of photonic techniques, which is their distinguishing quality. We reformulated part of the introduction to make this clear.

3. In line #117-199, the authors propose Miller's rule type estimate of susceptibility, but it is not clear to me if this is being used later or how it is relevant to the rest of the discussion. It is not even clear if such simplistic assumption is sensible in the present context. In the following paragraph, the authors propose a power law in reduced temperature to model the temperature dependence of the HHG and seemingly an exponent approximately equal to the (harmonic order)/2 works well for H3 and H7. Subsequently, the authors write, "This scaling matches the perturbative description of the nonlinear susceptibility" but this is perturbative in temperature and not electric field. Why does the nonlinearity depend on the reduced temperature in this fashion?

We thank the reviewer for the critical comments and understand that our explanation has not been sufficient to explain our intent. We have amended the text accordingly. In short, Miller's rule provides an empirical rule that allows the estimate of a material's nonlinear response based on knowledge of the linear response. Since we measure nonlinear responses, we used our model for the linear susceptibility to see whether such a rule holds and whether we can learn about the scaling of high harmonics as diagnostics of the nonlinear response or anisotropies.

Note that Miller's rule was derived for bound state susceptibilities. In contrast, high harmonic generation occurs via the optical field-driven electron wavepacket excursion. Thus, the process is nonperturbative and non-bound. Nevertheless, applying Miller's type rule is worthwhile in seeing whether and how the changed optical properties during the CDW impact HHG. Figure 1c shows that the optical conductivity at high H3, H5, and H7 are mostly temperature independent (with constant laser parameters). Thus, any change in harmonic yields should – with first approximation – occur according to Miller's rule due to a change of linear susceptibility at the fundamental frequency of 0.39 eV, taken to the order of the process.

We calculated the susceptibility at the fundamental frequency, and based on the measured reflectivity from Ref. 1, we found that the functional form of temperature-dependent susceptibility strongly depends on the scattering rate and differs from a simple order parameter approach. We agree with the reviewer that our initial assumption that the susceptibility directly follows the order parameter was too simple, e.g., the scattering rate plays a significant role.

Figure R3: Calculated (linear) susceptibility at 0.4 eV as function of temperature for different scattering rates. The curve in red is used for the following calculations with the Miller's rule.

Taking the simulated linear susceptibility from Fig. R3 to the respective order, Miller's rule-type approach approximates the harmonic responses with better agreement than the previous-order parameter scaling. We extract a scattering rate of 5 meV, corresponding with published values within 0.6-12 meV (Ref. 1) for the low-temperature range.

Figure R4: Measured (3rd, 5th and 7th) harmonic yield versus temperature for driving field orientations in (left) G-K and (right) both G-M directions. Red curves show the calculated susceptibility to the respective power, as Miller's rule type approximation in the perturbative regime. Such behavior agrees well for H3 and H7 as well as G-M1 direction of H5, meanwhile the other directions from H5 show anomalous behavior beyond the simple model.

Based on the reviewer's questions and remarks, we have extended our analysis and find that Miller's type scaling applies for harmonic orders in the perturbative (H3, H5) and nonperturbative regime (H7); see Fig R8. Taking the calculated susceptibility for different directions (G-K and G-M_{1,2}), we find that the harmonic yield deviates from the expected scaling for H5 for the direction G-K and G-M₂ but not for G-M₁; this is shown in R4 and now also added to the manuscript. While further experiments are needed to investigate the unexpected anisotropy between G-M₁ and G-M₂, we can already state that the change from perturbative to non-perturbative dynamics is not responsible for the observation. Further, a simple Miller's rule type scaling can be applied to predict the scaling of harmonic yield with temperature. Lastly, the dependencies of the harmonics allowed us to extract scattering rates in the CDW phase from the experimental measurement without the need to conduct broad frequency-range scans. We believe all of these findings are unexpected and valuable for further investigations.

4. The interpretation of the polarization analysis is perhaps the most problematic aspect of the paper. The asymmetry in the CDW wavevector has never been observed in any experiment as far as I am aware, and the authors do not cite any reports to that effect. While there is some disagreement regarding the origin of the periodic modulation and such, every momentum-resolved measurement, be it x-rays, neutron, STM or ARPES, have found a $2 \times 2 \times 2$ modulation and it is consistent with lattice dynamics studies through Raman and IR spectroscopies as well. Multiple theories have even explained this quantitatively and a summary of these results can be found in Ref. 40 for instance. It is therefore extremely unlikely that a deviation from the $2 \times 2 \times 2$ distortion would be missed in all of these measurements. Furthermore, because the $2 \times 2 \times 2$ superstructure is a commensurate CDW, a transition to unequal wavevector would involve a commensurate-incommensurate transition, which must be first order [McMillan PRB , 12(4), 1187 (1975)]. If such a transition exists between 80-100K as the authors posit, a hysteretic anomaly must appear in transport measurements. None of these contradictions are addressed in the paper, and in light of the overwhelming lack of supporting evidence, I find the proposed hypothesis highly implausible.

We respectfully disagree with the reviewer and refer to H. Kim et al. ⁴, a chiral effect of the CDW was observed, thus also reporting an anisotropy.

We apologize for any confusion between the parameters Q and ΔQ the model, as our text did not sufficiently highlight this. We have reworked these parts, especially in line 206.

Figure R5: Atom displacement in real space (orange arrows) and wave vector Q_i in the CDW phase of TiSe_2 . Black arrows indicate the corresponding orientation of the Q vectors. The atoms are Ti (black) Se1 (red) and Se2 (blue).

Q refers to the wave vector in the respective direction in reciprocal space, while ΔQ refers to the magnitude of the CDW order parameter, hence the name CDW strength, which is proportional to the displacement of atoms in real space. The triple Q CDW order arises from three independent real space displacements of each atom in the respective directions, as indicated in Fig. R5. A simple explanation for an anisotropy could be a strain, as illustrated in Fig. R6 (exaggerated displacements). Unequal displacement, however, does not break the commensuration of the material, as the crystal structure can still be reproduced by the same 2×2 supercell, indicated by the supercell in grey.

Figure R6: Left: Equal atom displacement δr_i in real space for commensurate CDW phase of TiSe_2 . Right: Unequal atom displacement in all three directions still results in a commensurate CDW formation. The 2×2 supercell is indicated in gray. Note: The displacement is exaggerated for visual illustration.

The reviewer refers to previous work with different experimental methods whose sensitivity may not be on par with the microscopic sensitivity of HHG. Note that we are observing a change of 1%, which may be hard to observe in ARPES measurements. Further, we refer to recent literature and the work of H. Kim et al. ⁴ in which they discuss the detection of a chiral CDW phase. Their and our results could also indicate band splitting along different M-directions in the CDW phase, an effect to which HHG spectroscopy is superbly sensitive.

5. Strain could be important here and it is certainly possible that the samples are getting strained at low temperatures due to some uncompensated thermal contractions. It is known that the CDW in TMDs and in TiSe_2 becomes incommensurate under pressure/strain/intercalation etc. However, in such a scenario, the authors should investigate quantitatively why H5 gets modulated while H3 and H7 appear to be less sensitive. Strong optical field itself can also strain the material, especially through piezo-optical coupling, but a comprehensive symmetry analysis is necessary to assess the feasibility of such process.

We respectfully disagree and point out that the reviewer's argument in the present question contradicts the previous argument. The reviewer states previously that "... because the $2 \times 2 \times 2$ superstructure is a commensurate CDW, a transition to unequal wavevector would involve a commensurate-incommensurate transition, which must be first order [McMillan PRB, 12(4), 1187 (1975)]. If such a transition exists between 80-100K as the authors posit, a hysteretic anomaly must appear in transport measurements." This must be observed in all harmonic orders, and this is in contradiction to our measurements. The new results shown in Fig. 4 show contradict the reviewer's point of reasoning, as the polarimetry analysis shows that H5 for G-M₂ only deviates but not for G-K and G-M₁. Further, the asymmetry is not observed for temperatures above the critical temperature (Fig. R4). The rotating field used in our polarimetry analysis also rules out that the strain is induced only in one direction by the field itself.

6. More importantly perhaps, it must be noted that similar asymmetry between HHG response for polarizations along symmetry-equivalent directions has been observed before in TMDs [Yoshikawa et al. Nat. Comm. 10, 3709 (2019)]. These measurements also found differences between zigzag and armchair directions, and all of these effects could be explained by band structure-related effects without invoking

strong correlation physics, CDW, or otherwise. The observations reported here could likely be a manifestation of similar phenomena.

We respectfully disagree with the reviewer. While the cited paper shows the sensitivity of HHG to the microscopic properties of a material, we show how the HH response changes with field orientation and temperature across the phase transitions. Our measurement provides unexpected new results, namely the scaling and sensitivity of HHG to the temperature-dependent changes in the material and the asymmetry of the behavior between G-K, G-M₁, and G-M₂. See also the new Method's section.

In typical harmonic spectroscopy the modulation can be modeled with the non-linear susceptibility tensors. Looking at the 5th order we can write the non-linear polarizability as $P_i^{(5)} \propto \sum_{ijklmn} \chi_{ijklmn}^{(5)} E_i E_j E_k E_l E_m E_n$ with $\chi_{ijklmn}^{(5)}$ being the 5th order susceptibility tensor and $A_0 \begin{pmatrix} \frac{\sqrt{2}}{2} \cos \theta & \sin \theta & \frac{\sqrt{2}}{2} \cos \theta \end{pmatrix}$ the electric field vector after projection for 45-degree incidence. For TiSe₂ having the point group #164 P-3m1 only 12 unique tensor components are non-zero. The exact form was obtained with TENSOR from the Bilbao Crystallographic Institute⁵. The harmonic yield is then proportional to the susceptibility and electric field as $Yield_N \propto \epsilon_0^2 |\chi^{(N)} E^N|^2$.

Figure R7: Left: Non-zero 5th order nonlinear susceptibility components as function of driving field polarization angle after projection. All components are symmetric around 90 degrees. Right: Measured H5 yield at 283 K, as function of the same angle and susceptibility-component fit. Out of 12 components only 2 produce a strict double peak feature with 4 more influencing the contrast at 90 degrees.

Figure R7 (left) shows all non-zero tensor components as a function of the driving field polarization angle. The right panel shows a fit to the H5 data at 283 K, where the shape is well reproduced from the only two components with one double lobe at 90 degrees (χ_{xxxzxx} and χ_{zxxzzy}). In this geometry, equivalent to our experimental 45-degree incidence, all tensor components are symmetric. This shows that the response of H5 should not be asymmetric, as observed in our experiment.

7. The authors allude to quantum phase transitions several times (abstract, introduction, conclusion) and goes on to write: “we demonstrate that HHG is exceptionally sensitive to symmetry breaking due to the quantum phase transitions and lattice dynamics”. However, a transition at 200K (or 80K) is most certainly not a quantum phase transition. Which transition are the authors alluding to?

We thank the referee for the feedback. We agree that the strict definition of a quantum phase transition is the abrupt change of the ground state of a system due to quantum fluctuations. We followed work in the literature, which called the thermally-driven CDW phase transition “quantum.” We have changed our wording accordingly.

8. The data along G-M direction has 3-4 points. Fitting it to a non-linear function with two (or more) parameters is highly questionable.

We respectfully disagree with the reviewer and not that if the fit would be a polynomial or unknown function, we would agree. Here, we are fitting a known functional form to the data. Thus, even the single data point at 83 K can directly differentiate if H5 has a minimum or follows the Miller’s rule scaling.

Overall, I do not find the results to be novel. Systematic study of HHG (up to 18th order) has been demonstrated in multiple TMDs over 5 years ago. Many aspects of the present dataset were also observed in the previous work including the non-monotonic harmonic intensities and polarization sensitivity. While the data might be technically sound, the authors fail to provide reliable evidence for their inferences. I do not recommend this manuscript for publication.

We are surprised at the inappropriate general statement of the reviewer, who simply compares our work to HHG in some TDMCs. Our work is not about showing that HHG can be obtained from a TMDC. Here, we successfully reveal the phase transition in TiSe_2 with HHG spectroscopy and measure scattering rates. We used the demonstrated extreme microscopic sensitivity of HHG and detected an asymmetry in the CDW phase, which manifests in the G- M_2 direction of H5 in the perturbative regime. A similar asymmetry has been found in tunneling spectroscopy and was linked to the chirality of the CDW phase⁴. Our observations are entirely novel and take the method of HHG polarimetry beyond previous work by investigating details of a phase transition with new information about microscopic properties of the nature of the CDW phase.

We thank the reviewer for highlighting some missing details and wording, which we have reworked. We hope that the improved manuscript, the additional analysis, and our response allow publication of our work.

Reviewer #2 (Remarks to the Author):

(...) The major outcome of this study is the demonstration of anisotropic 5th harmonic generation emission as a function of polarization in the charge-density-wave (CDW) phase. This finding could indicate band splitting along different M-directions in the CDW phase, a phenomenon that harmonic spectroscopy can probe with exceptional sensitivity.

The experimental results presented in the paper are of high quality. However, the discussion and interpretation could be improved to provide a clearer and more comprehensive presentation of the findings.

Harmonic spectroscopy is a novel approach, and its theoretical interpretation often relies on models where certain parameters are empirically adjusted to match experimental observations. This appears to be the case in this paper, although it is not explicitly stated. That said, high-quality experimental results like those presented here should not be undervalued simply due to the lack of direct methods for reproducing the findings or the absence of a definitive theoretical explanation. The intriguing physics observed through the experiment is compelling. Moreover, additional measurements have been conducted to rule out possible misalignments of the laser, further supporting the validity of the results. Given the novelty and importance of these experimental findings, particularly within the framework of this emerging spectroscopic approach, I believe the work is well-suited for publication in *Communications Materials*. However, the authors should first address the following concerns regarding the presentation and interpretation of the results, as well as clarify a few specific points.

We would like to thank the reviewer for assessing our work and providing valuable feedback. We sincerely appreciate the positive remarks and are delighted to read that the reviewer finds our reported method valuable, novel, and well-suited for publication. We took all of the reviewer's suggestions for improvement into account and made appropriate changes to the manuscript.

1. The manuscript states that “the experiment remained in the perturbative regime and moderate field amplitudes,” supported by a perturbative model that reproduces the temperature dependence of harmonics 3 and 7. However, when discussing harmonic 5 and in the conclusions, the authors mention a “deviation from simple perturbative dynamics.” Are we in the perturbative regime or not?

1a. A clear indication of a non-perturbative regime for harmonic 5 could have been provided by examining the harmonic intensity as a function of the driving field intensity. Was such an analysis attempted?

We thank the reviewer for raising this point. We did conduct an analysis of scaling of harmonic order with driving field intensity (pulse energy for constant beam size and pulse duration) to infer the regime. The result is shown in Fig. R8 below and we include this figure as Fig. S2 in the Supplement for completeness. We find that H3 and H5 scale perturbatively for the relevant energy / intensity range. H7 deviates from pure perturbative scaling.

Figure R8: Yield of harmonic orders 3, 5 and 7 as function of the driving field energy. Black lines are fits to extract the power-law for each harmonic. H3 and H5 agree well with the respective power laws in perturbation theory, while H7 has a significantly lower scaling.

1b. Perturbative dynamics imply that the highly non-linear regime characteristic of high-harmonic generation (HHG) is not accessed. Is this the case in this experiment? How many harmonics were observed (i.e., what is the cutoff)? Do the harmonics exhibit a plateau, or do they follow a power-law decay, characteristic of the perturbative regime?

We thank the reviewer for this question. We point out to the reviewer that the “highly-nonlinear” regime of HHG is not as nonlinear as one expects as the nonlinearity does not simply scale with harmonic order as one would expect from the perturbative regime. Further, the cutoff of HHG in materials in many times severely limited by the field strengths that can be applied without causing damage or a complete distortion of the physics one aims to observe. For instance, if a material is only to serve as host for HHG with the aim of light generation, then this is an entirely different regime of interaction as when we try to find a balance between getting many harmonic orders as spectroscopic signature of the bands whilst keeping their distortion minimal. Here our goal is the latter, thus we do not expect a large harmonic spectrum and also no plateau. Figure R8 shows a measurement in the energy range of our experiment which is bounded between the detection limit and the energy at which distortions and damage occurs. These limits are laser parameter and material dependent. Our measurement shows that H3 and H5 reasonably follow to the 3rd and 5th order, indicating the perturbative regime. Meanwhile, H7 shows significant deviation from the 7th order power law, hinting at its non-perturbative nature without showing saturation.

We further add with regard to the question about nonlinearity that the process in materials is due to electron hole recombination either within the first Brillouin zone or further away, as long as coherence is preserved. The long possible excursions result in an earlier onset of nonperturbative generation due to the stronger contributions from long trajectories. More meaningful than the pure order is a comparison of harmonic energy to the (field-free) bandgap. We find that H7 is the first above bandgap harmonic, further distinguishing it from H3 and H5. We include Fig. R8 as Fig. S2 in the Supplement.

1c. The deviation of harmonic 5 from harmonics 3 and 7 might not necessarily indicate non-perturbative behaviour. Instead, it could result from the “non-parametric” nature of harmonic 5. While the polarization may still be expressed as a Taylor expansion in susceptibility coefficients, these coefficients

are sensitive to resonances at specific frequencies and crystal-axis orientations. The discussion before the conclusion suggests that a resonance overlapping harmonic 5 could render the susceptibility coefficient complex, leading to a highly non-parametric harmonic generation. Can the authors clarify if this is the correct interpretation of their experimental findings?

We thank the reviewer for the chance to clarify and the reviewer's interpretation is spot on. We find that the H5 energy at 2 eV is close to the energy gap near the M point. Given that we established the perturbative nature of H5 within this experiment (see point 1b) we conclude that H5's anomalous behavior is likely a result of the harmonic generation near a resonance, and thus a potentially non-parametric process.

1. If it is confirmed that the experiment presents results of HHG in the perturbative regime, this should be clearly stated in the title, as HHG spectroscopy usually refers to the spectroscopy approaches that can be accessed from the non-perturbative description of the process. If the authors claim to be sensitive to highly non-linear processes, a better experimental evidence of this claim or support by theory should be provided.

We understand the intent of the reviewer's comment but historically for solids simply calling the investigation "harmonic spectroscopy" refers overwhelmingly to studies of 2nd and 3rd (and rarely 5th) orders in the perturbative regime. Numerous recent studies in solids, especially in fragile and exotic materials, claimed the term "high-harmonic spectroscopy" without pushing for particularly high harmonic orders or the harmonic cut-off, while still making heavy use of the perturbative dynamics and interpretations⁶⁻¹¹. Our experiment accesses both the perturbative and non-perturbative regimes dependent on harmonic orders, allowing us to test the limits of simple models like the Miller's rule or susceptibility tensors. Particularly H7 seems to agree well with the former (see Fig. R4, answer to referee 1) and not work well for the latter, with e.g. the simulation showing a non-sinusoidal angle dependence see Fig. R9. Further we observe non parametric behavior in H5, which is beyond the typical perturbative/non-perturbative debate.

Figure R9: Simulated intensity of H7 as function of the driving field polarization angle.

2. The manuscript claims "excellent agreement" between the mean-field model and experimental results. However, in the methods section, the interaction strength and order parameter are assigned different values along the translation vectors Q. Were these values obtained by fitting to the experimental data? If

so, this agreement stems from fitting rather than modeling, and this distinction should be explicitly stated.

We thank the referee for raising this point. In our theoretical modelling, we didn't fit any experimental data. The different interaction strength and order parameter along different CDW wave vector Q_i are introduced to mimic the anisotropic CDW order parameter strength possibly induced by the strain.

We apologize for any confusion between the parameters Q and ΔQ the model, as our text did not sufficiently highlight this. We have reworked these parts.

Figure R5: Atom displacement in real space (orange arrows) and wave vector Q_i in the CDW phase of TiSe_2 . Black arrows indicate the corresponding orientation of the Q vectors.

Q refers to the wave vector in the respective direction in reciprocal space, while ΔQ refers to the magnitude of the CDW order parameter, hence the name CDW strength, which is proportional to the displacement of atoms in real space. The triple Q CDW order arises from three independent real space displacements of each atom in the respective directions, as indicated in Fig. R5. A simple explanation for an anisotropy could be a strain, as illustrated in Fig. R6 (exaggerated displacements). Unequal displacement, however, does not break the commensuration of the material, as the crystal structure can still be reproduced by the same 2×2 supercell, indicated by the supercell in grey.

Figure R6: Left: Equal atom displacement δr_i in real space for commensurate CDW phase of TiSe_2 . Right: Unequal atom displacement in all three directions still results in a commensurate CDW formation. The 2×2 supercell is indicated in grey. Note: The displacement is exaggerated for visual illustration.

The corresponding values of U_{Q_i} , as phenomenological temperature-dependent parameters in our modelling, are obtained self-consistently to reproduce suitable strength of initial CDW order parameters, which in this case are set by hand around the isotropic value $\Delta(T) = \Delta_0 \sqrt{1 - (T/T_c)^2}$ with $\Delta_0 = 115$

meV to show the generic behavior of HHG under the anisotropic CDW order parameters rather than fitting. The obtained HHG spectra reproduces well the behavior of experimental observations, hence we claimed “excellent agreement” between the mean-field model and experimental results. We have toned this down and now claim very good agreement.

3. A possible chirality in the electronic states would be significantly coupled to the lattice [See PRB 3 L022003 (2021)]. The observed asymmetry between translation vectors Q introduced in the model reproduces the experimental results. Does this asymmetry correspond to a chiral point group in the crystal lattice? If so, chirality cannot be entirely ruled out as a source of the observed asymmetries.

The referee raises an interesting point about possible chirality within the CDW phase. As discussed in the previous point, we refer to the CDW strength ΔQ_i which is asymmetric in the three directions, leading to a different atom displacement in real space. Despite this, the structure is still reproduced by the same 2×2 supercell (see Fig.R6) and the Q vectors remain unchanged.

Regarding the main question, an asymmetric displacement can indeed lead to a chirality i.e. clockwise and anti-clockwise chiral domain formation which was reported (H. Kim et al. Nano Letters 24, 14323 (2024)) in a bi-layer TiSe_2 sample and earlier studies, both via scanning tunneling microscopy^{4,12}. Based on our and the cited findings, an enticing opportunity arises for future work to investigate the chiral nature of the CDW wave and whether such nm-sized chiral domains can lead to a chiral macroscopic response, despite the much larger active area from a beam size on the order of 100 microns.

For the current text we amended the section, mentioning the possible chirality of the sample while still showing no influence from slight ellipticity (or chirality) of the driving field as concluded from the simulation.

4. Before the conclusions, the authors say “Another explanation may lie in the backfolding of bands and their orbital character. While Fig. 1a illustrates the three conduction bands c_1 , c_2 , and c_3 along the Γ -M path, they split along the M directions, and only Γ -M3 has a band nearly resonant with H5”. It would be useful to show this split along the M directions. What M direction is shown in Fig.1(a)? What is the Γ -M3 direction?

Within the normal phase, the three Γ -M directions are symmetry equivalent, the bands are identical and they overlap. Therefore, the path Γ -M is shown only once. When the Brillouin zone is reduced in the CDW phase and all three bands are back-folded to the Γ point, the renormalization for each band is different. The indices for c_1 , c_2 and c_3 in the directions Γ -M₁ Γ -M₂ or Γ -M₃ were chosen based on convention and consistent with the definition of the unit cell. This description emerged from photoelectron spectroscopy measurements where these bands could be identified, but they are usually not shown as being split in their separate directions. We chose the same convention to provide a way to compare easily with existing literature.

We would like to keep the focus on the fact that the harmonics are probing of the v_1 and c_3 bands which are mainly participating within the three-step-model like picture for HHG in solids and their gap at the M points being nearly resonant to H5. To make this clearer we adjusted the quoted section within the main text. We agree with the reviewer that the notion for directions must be explained better and we added a figure to the supplementary material section. The identical new figure is shown here as Fig. R10.

Figure R10: The reciprocal space of TiSe₂. The dashed lines represent the reduced Brillouin zones folded by the CDW wave vectors Q_i . High symmetry points in the reduced and original Brillouin zone are denoted by symbols (Γ , M, K) with and without bar, respectively.

5. The manuscript states a peak intensity of 40 GW/cm². How was this value estimated? Does it refer to the field outside the sample or the transmitted field within it? Furthermore, simulations were performed at $A=0.12$ a.u., while the experimental field corresponds to $A=0.1$ a.u. Why was this discrepancy introduced?

The peak intensity is estimated from power, beam-profile and duration measurements which were conducted in tandem with the experiment. The quoted intensity is determined outside the sample since the refractive index of TiSe₂ changes with temperature. The experimental vacuum field strength for the polarization scans is 0.078 ± 0.007 V/Å, which would correspond to a field strength inside the TiSe₂ of roughly 0.032 ± 0.003 V/Å at 300 K and change as the sample is cooled.

We note that the quoted field strengths are not in atomic units. Instead the simulation uses the vector field strength in dimensionless units, which is calculated as follows from the experimental value: $A_0 = \frac{eE_0a}{\hbar\omega} = \frac{e \times 0.078 \frac{V}{\text{Å}} \times 3.54 \text{ Å}}{0.4 \text{ eV}} \approx 0.69$ For the simulation $A_0 = 1.2$ is used, being within the same order of magnitude. There exists no discrepancy between values in the experiment and simulation. For clarity in the methods-section we now include the definition for A_0 and show the experimental vacuum field strength.

6. Rotating the polarization, instead of the sample, means that for different polarization angles, one must consider that s and p components behave differently at the sample/vacuum interface due to different Fresnel transmission coefficients for the driving field and reflection coefficients for the harmonics. This effect is especially relevant at a 45° incidence angle and could explain the near-zero HHG at 0° and 180°, where s-polarized driving fields might experience suppressed transmission. Polarization-resolved HHG is better performed when the sample is rotated instead of the driving field. I understand this may be impractical when using a cryostat for temperature-dependent measurements. However, this limitation, the related consequences, and the experimental signatures related to this geometry should be better explained. Moreover, it is not clearly stated if the model is taking into account this particular geometry.

The reviewer is of course correct and we have taken the projections of the optical field during rotation into account. We also added a sketch of the experimental geometry in Fig. 1e to further improve clarity, it is shown here in Fig. R11.

Figure R11: Illustration of the reflection geometry in the experiment. The incidence angle α_{in} is 45-degrees, while theta refers to the driving field's polarization and α to the crystal lattice offset rotated within the x-z plane. The 1st Brillouin zone is also shown with the corresponding symmetry directions. Drawn in blue is the simulated angle dependent H5 response without projections.

We show in Fig. R12 for completeness how H5 depends on the polarization angle for "normal" and 45° incidence.

Figure R12: Harmonic 5 intensity as function of polarization rotation for normal 0° incidence (a) or 45° incidence (b) projection to the experimental geometry.

While being beyond the scope of this work the acquired harmonic signal from such geometry will contain additional crystal information i.e. the z-component when the field is p-polarized, probing the often neglected out of plane component. To address the point of mechanical sample rotation, complications will arise from more than just impracticalities of the cryostat. Polarization rotation is significantly more reliable compared to mechanical sample rotation, due to the following factors:

- The sensitivity of the reflection geometry to minimal changes in alignment, which is especially true for the nonlinear processes during HHG.
- The centering problem, where the ~1 mm small sample will be hit at different positions during the rotation.

-And finally, the possibility that the CDW phase is not uniform along the sample surface or forms different domains as found by ^{4,12}, making any changes in harmonic yield during a rotation scan unreliable.

7. The experimental results are presented only for angles ranging from 0° to 180°. Why was a full 360° scan not performed?

The applied optical field was linearly polarized, which means that only half of the angular space has to be sampled. A scan from 0 to 180 degree is equivalent to a scan from 180 to 360 degrees. To optimize for accuracy against measurement time we opted to perform a high number of short scans as compared to a low number of long scans where potential long-term drift in the laser power or cryostat temperature could create a systematic error. The shown data comes from averaging 10-15 repeated scans.

References

1. Li, G. *et al.* Semimetal-to-semimetal charge density wave Transition in 1T-TiSe₂. *Phys Rev Lett* **99**, 2–5 (2007).
2. Timusk, T. & Tanner, D. B. Timusk-Tanner-Ginsberg-1989.pdf. *Physical Properties of High-Temperature Superconductors I* (1989).
3. Monney, C. *et al.* Mapping of electron-hole excitations in the charge-density-wave system 1T-TiSe₂ using resonant inelastic x-ray scattering. *Phys Rev Lett* **109**, (2012).
4. Kim, H., Jin, K.-H. & Yeom, H. W. Electronically Seamless Domain Wall of Chiral Charge Density Wave in 1 T -TiSe₂. *Nano Lett* **24**, 14323–14328 (2024).
5. Gallego, S. V., Etxebarria, J., Elcoro, L., Tasci, E. S. & Perez-Mato, J. M. Automatic calculation of symmetry-adapted tensors in magnetic and non-magnetic materials: a new tool of the Bilbao Crystallographic Server. *Acta Crystallogr A Found Adv* **75**, 438–447 (2019).
6. Han, S. *et al.* Extraction of higher-order nonlinear electronic response in solids using high harmonic generation. *Nat Commun* **10**, 1–6 (2019).
7. Klemke, N. *et al.* Polarization-state-resolved high-harmonic spectroscopy of solids. *Nat Commun* **10**, 1319 (2019).
8. Alcalà, J. *et al.* High-harmonic spectroscopy of quantum phase transitions in a high-T_c superconductor. *Proceedings of the National Academy of Sciences* **119**, 1–6 (2022).
9. Kaassamani, S. *et al.* Polarization spectroscopy of high-order harmonic generation in gallium arsenide. *Opt Express* **30**, 40531 (2022).
10. Takeda, K. S. *et al.* Ultrafast Electron-Electron Scattering in Metallic Phase of 2 H-NbSe₂ Probed by High Harmonic Generation. *Phys Rev Lett* **132**, (2024).

11. Katayama, I. *et al.* Three-dimensional bonding anisotropy of bulk hexagonal metal titanium demonstrated by high harmonic generation. *Commun Phys* **7**, 1–8 (2024).
12. Ishioka, J. *et al.* Charge-parity symmetry observed through Friedel oscillations in chiral charge-density waves. *Phys Rev B Condens Matter Mater Phys* **84**, (2011).

We would like to thank the reviewers for their careful consideration of our work, and we appreciate their valuable feedback, which has tremendously helped to improve the manuscript

Reviewer #1 (Remarks to the Author):

I would like to thank the authors for their reply. The newly added references, calculations and revisions in the resubmitted manuscript certainly clarifies a number of important points which I found to be rather confusing in the previous draft.

The calculations of the harmonic yield based on the linear susceptibility and corresponding Miller's rule estimates are more concrete compared to the previous iteration. It is interesting to see the importance of the scattering rate in this regard, although it is perhaps not surprising. The point group symmetry analysis and revisions to Fig. 2 certainly make for a more compelling case for the anomalies associated with the 5th harmonics response.

Few general remarks:

1. Ref. 36 in the main text is probably incorrect. Based on Fig. 1 caption I was expecting to find the PRL (2007) from Li et. al. but the current manuscript on my end is referring to Boyd's nonlinear optics textbook instead.

We thank the reviewer for the comment. The bibliography now states the correct reference.

2. It might be useful to add the Drude-Lorentz parameters to the supplemental. Temperature dependence of the scattering rate for the 0.4eV Lorentzian is certainly useful.

We have included this now in the Supplement. The fit parameters are now provided as a table.

I am satisfied with the revised manuscript, and I would recommend the manuscript for publication.

We want to thank the reviewer for the nice and encouraging statement and the favorable judgment of our work. It helped elevate our methodology and understanding, leading to an overall improvement in the quality of this work. We addressed the final points within this document and amended the manuscript to correct minor mistakes.

Reviewer #2 (Remarks to the Author):

The authors have addressed most of the points raised in my previous review. I have only a couple of comments regarding the answers to my questions.

In answering comment 4, the authors say: “We would like to keep the focus on the fact that the harmonics are probing the v_1 and c_3 bands, which are mainly participating within the three-step-model-like picture for HHG in solids and their gap at the M points being nearly resonant to H5.” But then, in the new version of the paper, the authors now state: “Previous broadband studies (Fig. 1c) show that the optical response at harmonic frequencies is nearly independent of temperature. This suggests that harmonic generation is primarily influenced by changes at the fundamental frequency, where the resonance appears.” This contradicts the previous statement that, being H5 close to a resonance, we must be highly sensitive to H5. Being $\chi(5\omega_0)$ at resonance, harmonic generation can be primarily influenced by changes of the χ at the harmonic frequency (not only the fundamental). Fig. 1c only demonstrates that it is not dependent on the change in the conductivity. Please clarify.

We apologize for our previous response, which caused additional confusion. As stated, the carrier excitation occurs at the Γ point (v_1 to c_3). This bandgap changes significantly with temperature. On the other hand, the (new) M point is close to resonance with H5, and its bandgap should not change considerably below T_c . This agrees with the conductivity curves in Fig. 1c. For clarity, we also point out that $\chi^{(5)}(\omega_0)$ and $\chi^{(1)}(5\omega_0)$ are distinct. Being close to resonance allows for a high probability of recombination between electrons and holes near the M point, and the H5 generation can be well understood as a process similar to an interband three-step model. In contrast, H3 would have a significant intraband contribution, with higher energy bands like H7 playing a role. Therefore, we expect H5 to be the most direct probe of the electron-hole trajectories along the band structure, providing an intuitive understanding of its sensitivity to directions such as Γ -M and Γ -K. We hope this explanation clarifies our point.

Comment 5. I thank the authors for the clarification and the amendments. I do not agree that $A_o = 0.69$ and $A_o = 1.2$ are equivalent for HHG. A change of almost two times in the field strength dramatically changes the result of simulations for HHG, even in the perturbative regime. I ask again to clarify in the paper why, in the simulation, a value close to the one measured was not used. What would be the result of the simulation by using this value?

We thank the reviewer for raising this point. We want to clarify that our numerical simulation aims to reproduce the key features observed in the experiment only qualitatively and to identify the dominant underlying physical mechanism.

While the field strength between the experimental and theoretical values differs, we do not consider this discrepancy to significantly impact our conclusions. We have verified that the qualitative behavior remains robust across a range of field strengths, specifically within half an order of magnitude. Therefore, a factor of 2 is not of real significance and does not change the statement. Notably, the chosen theoretical field strength produces a modulation contrast in H5 that more closely matches the experimental data, which supports the relevance of our parameter choice.

We also emphasize that modeling such a correlated material with high quantitative accuracy is very complex. Therefore, we employed a simplified phenomenological mean-field model to capture the essential physics of charge density wave (CDW) orders. The simplified nature of this model may naturally lead to some differences in specific simulation parameters, such as the field strength, compared to the real material.

Nevertheless, the values used are within the same order of magnitude. More importantly, the observed asymmetry in the high harmonic generation (HHG) spectrum and the emergence of anisotropic CDW order parameters are qualitatively consistent with the experiment. These central findings are not sensitive to the exact value of the field strength as long as the primary features of the HHG spectrum are well reproduced, which is the case here.

Check references in the new version of the manuscript, some have not been updated. I've found one in the methods section "The parameters for the tight-binding model can be found in Ref. 40" but now Ref. 40 is not anymore the correct reference.

We thank the reviewer for pointing out the mistake. We updated the reference.

Final remarks:

I will start from one of the most important points raised by the first referee to elaborate my final assessment. The referee comments about the similar asymmetry between HHG response observed in TMDs [Yoshikawa et al., Nat. Commun. 10, 3709 (2019)]. I would like to add to the discussion that Yoshikawa et al. point to a different type of asymmetry (zigzag horizontal vs. armchair vertical). No change is observed or claimed between the horizontal and 60° directions, which are (and must be) completely equivalent directions in the absence of a change in the structure.

The new paragraph introduced by the authors, which includes a fit with the susceptibility tensor parameters, nicely shows that. The symmetry of TiSe₂ in the standard phase enforces the M1 and

M2 directions to be equivalent. The high sensitivity of HHG to the structure and the reported HHG response as a function of the temperature strongly point to the fact that the effect observed is associated with the formation of the superlattice.

I, however, agree with the first referee that whether the specific effect observed is due to the backfolding of the bands in the low-temperature phase, the anisotropy of the CDW, or other sources of symmetry breaking associated with the phase transition, is still a matter of discussion. But in any case, I believe the authors have provided enough evidence that it must be related to the CDW phase.

That said, the theoretical results presented in the paper give sufficient indication that the CDW anisotropy can show up in HHG with quite a strong signature, and the good agreement with the experiment (even if some parameters are arbitrarily chosen) points to the fact that the explanation provided by the authors is reasonable.

I think these results merit publication in this journal.

I confirm my previous evaluation, and I believe the work is well-suited for publication in Communications Materials after addressing the minor comments above.

We want to thank the reviewer for the nice and encouraging statement and the favorable judgment of our work. It helped elevate our methodology and understanding, leading to an overall improvement in the quality of this work. We addressed the final points within this document and amended the manuscript to correct minor mistakes.

Communications Materials is committed to improving transparency in authorship. As part of our efforts in this direction, we are now requesting that all authors identified as 'corresponding author' create and link their Open Researcher and Contributor Identifier (ORCID) with their account on the Manuscript Tracking System prior to acceptance. ORCID helps the scientific community achieve unambiguous attribution of all scholarly contributions. You can create and link your ORCID from the home page of the Manuscript Tracking System by clicking on 'Modify my Springer Nature account' and following the instructions in the link below. Please also inform all co-authors that they can add their ORCID to their accounts and that they must do so prior to acceptance.

If you experience problems in linking your ORCID, please contact the Platform Support Helpdesk.

This email has been sent through the Springer Nature Tracking System NY-610A-NPG&MTS

Confidentiality Statement:

This e-mail is confidential and subject to copyright. Any unauthorised use or disclosure of its contents is prohibited. If you have received this email in error please notify our Manuscript Tracking System Helpdesk team at <http://platformsupport.nature.com> .

Details of the confidentiality and pre-publicity policy may be found here <http://www.nature.com/authors/policies/confidentiality.html>

Privacy Policy | Update Profile